# TROPOMI/S5P Total Column Water Vapor Validation against AERONET ground-based measurements

Katerina Garane[1], Ka Lok Chan[2,3], Maria-Elissavet Koukouli[1], Diego Loyola[2], Dimitris Balis[1]

[1]Laboratory of Atmospheric Physics (LAP), Aristotle University of Thessaloniki (AUTH), 54124 Thessaloniki, Greece
[2]Deutsches Zentrum für Luft und Raumfahrt (DLR), Institut für Methodik der Fernerkundung (IMF), 82234 Oberpfaffenhofen, Germany
[3]Rutherford Appleton Laboratory Space, Harwell Oxford, United Kingdom

*Correspondence to*: Diego Loyola (Diego.Loyola@dlr.de)

**Abstract.** Water vapor plays an important role on the greenhouse effect, rendering it an atmospheric constituent that requires
continuous and global monitoring by different types of remote sensing instruments. The TROPOMI/S5P Total Column Water Vapor (TCWV) is a new product retrieved from the visible blue spectral range (435–455nm), using an algorithm that was originally developed for the GOME-2/MetOp sensors. For the purposes of this work, 2.5 years of continuous satellite observations at high spatial resolution are validated against co-located (in space and in time) precipitable water Level 2.0 (quality-assured) ground-based measurements from the NASA AERONET (AErosol RObotic NETwork). The network uses
CIMEL sunphotometers located at approximately 1300 stations globally to monitor precipitable water among other products. Based on data availability, 369 of the stations were used in this study. The two datasets, satellite and ground-based, were co-located and the relative differences of the comparisons were calculated and statistically analyzed. The Pearson correlation coefficient of the two products is found to be 0.91 and the mean bias of the overall relative percentage differences is of the order of -2.7 %. For the Northern Hemisphere mid-latitudes (30°N- 60°N), where the density of the ground-based stations is
high, the mean relative bias was found to be -1.8 %, while in the tropics (±15°) the TROPOMI TCWV product has a relative dry bias of up to -10 %. The effect of various algorithm and geophysical parameters, such as air mass factor, solar zenith angle, clouds and albedo are also presented and discussed. It was found that the cloud properties affect the validation results, leading the TCWV to a dry bias of -20 % for low cloud heights (CTP > 800hPa). Moreover, cloud albedo introduces a wet bias of 15 % when it is below 0.3 and a dry bias up to -25 % when the clouds are more reflective. Overall, the TROPOMI/S5P TCWV
product, on a global scale and for moderate albedo and cloudiness, agrees well at -2.7 ± 4.9 % to the AERONET observations, but probably within about -8 to -13% with respect to the "truth".

## 1 Introduction

The greenhouse effect, i.e., the infrared radiation energy trapped within the Earth-atmosphere system by atmospheric gases and clouds, is found to be highly dependent on the amount of water vapor in the atmosphere (Raval et al., 1989). Water vapor
is a natural greenhouse gas that originates from the evaporation of the Earth's water and absorbs the heat radiated by the Earth.

It is transported by the atmospheric circulation and part of the water vapor follows a cycle that consists of cloud formation via condensation, transportation and return to the Earth's surface by precipitation, as rain or snow. It has a major positive feedback, ranging from 1.1 $Wm^{-2}K^{-1}$ to 2.4 $Wm^{-2}K^{-1}$, with a mean value of 1.7 $Wm^{-2}K^{-1}$, hence its effect on global warming can be double the $CO_2$ contribution (Colman, 2003). The way that water vapor affects the climate's energy balance is described,

among others, by Inamdar and Ramanathan (1998): following the warming of the Earth's surface and troposphere by the increasing levels of $CO_2$ and other greenhouse gases, the water vapor content of the atmosphere also increases and further contributes to the greenhouse effect, hence to the atmosphere's warming. Therefore, water vapor strongly determines the atmosphere's response to surface warming. Nevertheless, under certain circumstances it is conceivable that negative feedback could result from the increase of the water vapor content, hence an increase in cloudiness that could lead to the cooling of the

atmosphere. Furthermore, the stratospheric water vapor load is significantly determined by methane and its oxidation within the stratosphere (Le Texier et al., 1988; Oman et al., 2008). It is evident that the net effect that water vapor changes can have on the climate is not clear yet. Additionally, water vapor as a chemical compound has another crucial role in the atmosphere since it is the origin of the tropospheric hydroxyl radical, which is a significant oxidant in the troposphere, and affects the ozone depletion in the stratosphere over high-latitude areas (Dlugokencky et al., 2016).

Being such an important factor for the evolution of the greenhouse effect and the projection of future climate change, water vapor is an atmospheric constituent that requires continuous and global monitoring by different types of remote sensing instruments and in individual spectral bands, such as microwave, short wave infrared and visible bands. We mention here the space-born Medium Resolution Imaging Spectrometer (MERIS) retrievals in the near-infrared (NIR) over land surfaces and coastal areas with the Special Sensor Microwave Imager (SSM/I) TCWV retrievals in the microwave spectra over ocean

surfaces (Lindstrot et al., 2014); the TCWV retrieval in the visible blue spectral band for the Global Ozone Monitoring Experience 2 (GOME-2) instruments on board the European Organization for the Exploitation of Meteorological Satellites (EUMETSAT) MetOp satellites (Chan et al., 2020); the EOS Aura Microwave Limb Sounder (MLS) for water vapor product (EOS, 2017); the MODIS (Moderate-resolution Imaging Spectroradiometer) on board Terra and Aqua total column water vapor (Diedrich et al., 2015); the Japanese Space Agency Greenhouse Gases Observing SATellite (GOSAT) column-averaged

dry-air mole fraction water vapor (Dupuy et al., 2016), etc. Furthermore, long-term ground-based observations also exist such as by the Total Carbon Column Observing Network (TCCON) of ground-based, high-spectral-resolution Fourier Transform Spectroscopy instruments (Wunch et al., 2011); by the ground-based Global Navigation Satellite System, GNSS (Gendt et al., 2004); by the GCOS Upper Air Network, GUAN, radiosondes (Turner et al., 2003) and by the Aerosol Robotic Network, AERONET, sun photometers (Pérez-Ramírez et al., 2014).

The TROPOMI/S5P Total Column Water Vapor (TCWV) is a new global product retrieved from the blue wavelength band (435–455nm). The retrieval algorithm was developed by the German Aerospace Center (DLR) within the framework of the European Space Agency's (ESA) Sentinel 5 Precursor Product Algorithm Laboratory (S5P-PAL), using as a basis the algorithm that was originally developed for the GOME-2 (Global Ozone Monitoring Experiment-2) TCWV products. The GOME-2

algorithm (Chan et al., 2020) was adjusted for the TROPOMI/S5P instrument in terms of spectral analysis, updated air mass factor calculations and a new surface albedo retrieval approach and is comprehensively described in Chan et al. (2022).

Borger et al. (2020) also retrieved TCWV from the same spectral band of TROPOMI/S5P measurements using the two-step Differential Optical Absorption Spectroscopy (DOAS) approach. The product was intercompared to the Special Sensor Microwave Image/Sounder (SSMIS) onboard NOAA's f16 and f17, the European Centre for Medium-Range Weather Forecasts (ECMWF) reanalysis model ERA-5 TCWV data and ground-based GPS data from the SuomiNet network. It was found that over ocean and under clear-sky conditions the retrieved TROPOMI/S5P TCWV captures well the global water vapor distribution. Over land, the retrieved TCWV was found to be underestimated by about 10 %, especially during boreal summer, which was attributed to the uncertainty of the external input data, hence some recommendations are given for the use of the product (effective cloud fraction <20 % and AMF>0.1). The methods of Borger et al. (2020) and Chan et al. (2022), are similar in principal but they differ in some important aspects such as: (i) Chan et al. (2022) fit for the 435-455nm spectral range, while Borger et al. (2020) use a slightly different wavelength range, 430-450nm; (ii) for the AMF calculation, the algorithm of Borger et al. (2020) assumes an exponential decay profile with empirical parameterization of the water vapor scale height, while Chan et al. (2020) use an a-priori profile from the statistical analysis of historical data and they dynamically pick the most appropriate one; (iii) for the surface albedo parameter, Chan et al. (2020) use the TROPOMI/S5P GE_LER which is derived at the same spectral fitting range (435-455nm), while Borger et al. (2020) use the OMI surface albedo retrieved at 442nm. The comparison of the two surface albedo products is extensively discussed in Chan et al. (2022).

Schneider et al. (2020) also introduced the retrieval of a clear-sky TCWV product retrieved from a different TROPOMI/S5P wavelength band, namely from its short-wave infrared (2305-2385 nm) observations. The product retrieval was further developed by Schneider et al. (2022) to also cover cloudy scenes and was validated against co-located ground-based Fourier transform infrared (FTIR) observations by the Total Carbon Column Observing Network (TCCON). The validation results showed that under clear-sky conditions the satellite product has a 2.9 % bias with respect to TCCON, which becomes 11 % for cloudy scenes. Compared to Chan et al. (2022), the Schneider et al. (2020, 2022) algorithm employed a completely different technique and due to the differences in the spectral range of the measurement, the final water vapor product has a different vertical sensitivity.

Another TCWV product retrieved by the Air-Mass-Corrected DOAS (AMC-DOAS) scheme based on TROPOMI/S5P data in the spectral area 688 to 700 nm, was presented by Küchler et al. (2021). After the retrieval, the product was post-processed to correct for surface albedo, cloud and across-track features. It was compared to ECMWF ERA-5, SSMIS data and the two scientific S5P/TROPOMI TCWV products that were mentioned above, i.e. the TCWV products described and validated by Borger et al. (2020) and Schneider et al., (2020 and 2022). These comparisons showed that over sea, AMC-DOAS underestimates TCWV with respect to ERA-5 TCWV by about $2\,\mathrm{kg\,m^{-2}}$, while its agreement to the TROPOMI/S5P TCWV from Borger et al. (2020) is within $1\,\mathrm{kg\,m^{-2}}$ over both land and ocean. Finally, with respect to the TCWV from Schneider et al. (2020 and 2022), averaged differences of around $1.2\,\mathrm{kg\,m^{-2}}$ were found.

The objective of this work is to validate the TROPOMI/S5P TCWV product retrieved from the blue band from the algorithm that was developed by DLR (Chan et al., 2022). For our validation purposes, the co-located precipitable water Level 2.0 (cloud screened, quality-assured and calibrated) ground-based measurements from the NASA AERONET (https://aeronet.gsfc.nasa.gov/, AErosol RObotic NETwork; Giles et al., 2019), were used. The network uses CIMEL spectral Sun photometers, which are automatic, solar powered and self-calibrating instruments that robotically scan the sun and the sky and measure atmospheric aerosol optical properties and precipitable water (Holben et al., 1998). The AERONET database provides precipitable water observations at approximately 1300 stations globally.

In Sect. 2, the characteristics of the available satellite and ground-based data used in this work are given. Sect. 3 describes the co-location methodology as well as the ground-based dataset quality control protocols. Sect. 4 presents the global validation results of TROPOMI/S5P TCWV and a discussion about the dependence of the satellite product on various parameters. Finally, a summary and the conclusions are given in Sect. 5.

## 2 Data sources

### 2.1 TROPOMI/S5P total column water vapor

The TROPOspheric Monitoring Instrument (TROPOMI, http://www.tropomi.eu/) on board the Copernicus Sentinel-5 Precursor (S5P) was launched in October 2017, monitoring the Earth's atmosphere using four spectrometers with spectral bands in the ultraviolet (UV), the visible (UVIS), the near-infrared (NIR) and the shortwave infrared (SWIR) wavelengths (Veefkind et al., 2012). The observations are performed in a sun-synchronous low-Earth orbit with a local equatorial crossing time of 13:30 LT and daily global coverage with 14 orbits per day. Its spatial resolution was 3.5 km (across-track) by 7.0 km (along track) up to 6th August 2019, when it was modified to 3.5 km (across-track) by 5.5 km (along track). Its swath width is 2600 km, consisting of 450 ground pixels across-track, which provides daily global coverage. The TROPOMI instrument and its pre-launch calibration techniques are thoroughly described by Kleipool et al. (2018), while the in-flight calibration is analyzed in Ludewig et al. (2020).

The TROPOMI/S5P Total Column Water Vapor (TCWV) is a new product retrieved from the sensor's observations in the visible blue band (435–455nm). The retrieval algorithm, thoroughly described in Chan et al. (2022), is based on the GOME-2 TCWV algorithm (Chan et al., 2020), which is utilizing the Differential Optical Absorption Spectroscopy (DOAS) technique (Platt and Stutz, 2008). In short, a two steps approach is followed: first retrieving slant columns through the spectral analysis of the TROPOMI measurements in the blue band, and then converting the slant columns to vertical columns using an iterative air mass factor (AMF) calculation. Compared to the GOME-2 algorithm, some improvements were applied concerning the spectral retrieval, the air mass factor calculations and the surface albedo input parameter, for which the GE_LER (Geometry-dependent effective Lambertian equivalent reflectivity) that is produced by TROPOMI (Loyola et al., 2020), is used. Finally, the cloud information (e.g., cloud fraction, cloud top pressure and cloud albedo) are taken from the TROPOMI operational cloud product (Loyola et al., 2018). According to Chan et al. (2022), TROPOMI/S5P reports lower TCWV values by 1.24

kg/m$^2$ over land compared to ERA-5 TCWV reanalysis data, and by 1.74 kg/m$^2$ with respect to GOME-2 observations.

Additionally, they report that the uncertainty of TCWV observations over the tropics is 10-19 % under clear skies (effective cloud fraction < 0.5). The TCWV products from TROPOMI/S5P and GOME-2 were also validated against GNSS data from 235 European stations, by Vaquero-Martinez et al. (2022). They found that the correlation coefficient of the scatter plot comparing TROPOMI/S5P to GNSS TCWV co-located data is 0.93 and showed that TROPOMI underestimates TCWV by about −3 % for water vapor content above 10 kg/m$^2$.

For this work, 2.5 years (May 2018 to December 2020) of continuous TCWV satellite observations were made available. The dataset was filtered according to Chan et al. (2022), following the criteria: (a) solar zenith angle <85°, (b) effective cloud fraction <0.5, (c) Root Mean Square fit residual <0.002 and (d) Air Mass Factor >0.1. Figure 1 shows four seasonal global maps of the TROPOMI/S5P TCWV: panel (a) depicts December to February; panel (b) March to May; panel (c) June to August and panel (c) September to November. Throughout the year, the tropics hold the higher TCWV content, up to 80 kg/m$^2$

occurring mainly during summer and autumn. Over land in the Northern Hemisphere, where most of the ground-based stations are located, the TCWV is below 50 kg/m$^2$, decreasing to less than 5-10 kg/m$^2$ closer to the poles.

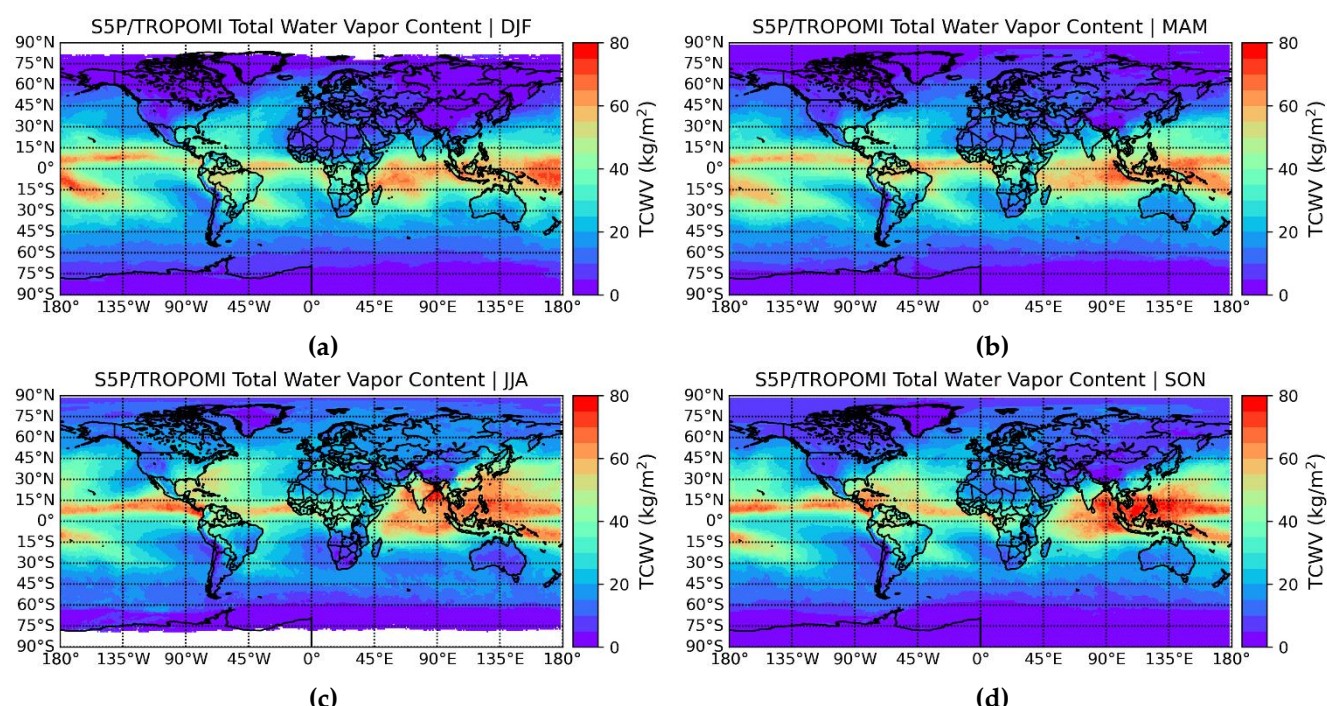

**Figure 1: Seasonal global maps of the TROPOMI/S5P TCWV product (in kg/m$^2$). Panel (a): December – February; panel (b): March – May; panel (c): June – August and panel (d): September – November**

## 2.2 Ground-based observations

The database used as ground-truth for the S5P TCWV validation consists of archived Cimel precipitable water observations that were downloaded from the AERONET website (https://aeronet.gsfc.nasa.gov/). The network uses Cimel Sun photometers located at about 1300 stations globally to monitor precipitable water, among other products, every 15 minutes. The Cimel instruments perform direct sun measurements when the optical path between the instrument and the sun is cloud-free. The AERONET processing algorithm was presented by Smirnov et al., (2004). Currently, Version 3 (Giles et al., 2019) of the algorithm is used for the retrieval and it is stated within the archived data files that "the data are automatically cloud cleared and quality assured with pre-field and post-field calibration applied". AERONET data are provided in three quality levels, namely 1.0, 1.5 and 2.0 (https://aeronet.gsfc.nasa.gov/new_web/aot_levels_versions.html):

- Level 1.0 data use the pre-field deployment sun calibration.
- Level 1.5 data use Level 1.0 data and apply a cloud-screening and automatic quality control procedures.
- Data are raised to Level 2.0 after applying the final post-field deployment sun calibration to Level 1.5 data.

Here, Level 2 precipitable water observations were used to achieve the best possible quality for the ground-truth.

The AERONET dataset covers about 25-30 years of measurements, depending on the station, and it was extensively used for the MODIS water vapor product validation (Bennouna et al., 2013; Diedrich et al., 2015; Bright et al., 2018; Shi et al., 2018; Martins et al., 2019). Schneider et al. (2010) found that Cimel instruments have a clear sky dry bias, which is larger in winter (25.5 %), decreasing during spring (11.5 %), and becomes a minor wet bias (2 %) in the summer months. The seasonality in the dry bias of the Cimel observations is caused by their restriction to clear-sky measurements. The AERONET precipitable water vapor product was evaluated by Pérez-Ramírez et al. (2014), where it was compared to water vapor retrievals from radiosonde observations and other ground-based retrieval techniques, such as microwave radiometry (MWR) and GPS for a few sites. It was found that the AERONET precipitable water has a dry bias of approximately 5–6 % in the retrievals and a total estimated uncertainty of 12–15 %. Weaver et al. (2017) also intercompared water vapor measurements performed by different types of instruments, namely radiosondes, sunphotometers, FTIR spectrometers and a microwave radiometer, at the Eureka, Nunavut site. They showed that the sunphotometers operated at two nearby sites report lower water vapor observations compared to co-located FTIR or Atmospheric Emitted Radiance Interferometer (AERI) instruments, by 15 % or 3.3 %, respectively. An approximate mean bias for the AERONET TCWV product that results from all these studies, that are based on various stations, temporal coverages and reference measurements, is -5 to -10 %.

Campanelli et al. (2018) validated precipitable water vapor content from ESR/SKYNET radiometers against GNSS/GPS and AERONET over three different sites in Europe and found that the agreement was within the reported uncertainties. The total uncertainty of sun photometer retrievals was estimated to be less than 10 % (Smirnov et al., 2004; Alexandrov et al., 2009; Pérez-Ramírez et al., 2014). According to Martins et al. (2019), this percentage was expected to be improved with the implementation of the version 3 of the retrieval algorithm (Giles et al., 2019).

The extended network of automatic and quality-controlled observations provides very dense (spatially and temporally) coverage of North and South America, Europe, South-East Asia, as well as Western Africa. This fact, in addition to the use of a single standardized retrieval algorithm and the consistency in instruments' calibration, are strong advantages in favor of
180 using the AERONET for this validation work.

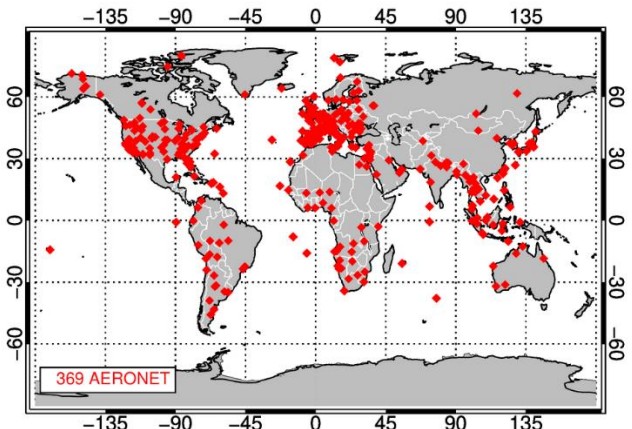

**Figure 2: Spatial distribution of the 369 AERONET ground-based stations used for the comparisons to TROPOMI/S5P TCWV product.**

The data files retrieved from AERONET are available in ASCII format in daily, monthly or instantaneous (i.e. measurements performed every 15') temporal analysis. Here, the instantaneous precipitable water observations were used, for the time period May 2018 to December 2020, depending on the availability of data for each individual station. Out of the 1304 stations, only 596 reported Level 2 precipitable water measurements after 2017. An in-house quality control based on the visual and statistical analysis of the available datasets per station, ensured that only stations with data that cover fully the time period of our study,
or cover at least 20 out of the 32 months of the TROPOMI/S5P dataset, are contributing to the ground-based reference dataset. As a result, the final number of stations to be used for the validation of TROPOMI/S5P TCWV was reduced to 369. Figure 2 shows their geographical distribution.

**3. Co-location methodology and AERONET stations quality control**

As a first step in the analysis, a dataset of overpass files was created whereupon all TROPOMI/S5P pixels within a 10 km
radius from the AERONET stations were extracted from the original orbital files. The use of the 10 km radius was based on the high spatial resolution of TROPOMI/S5P observations (3.5 x 7 km$^2$ until August 2019 and 3.5 x 5.5 km$^2$ thereafter). Moreover, other studies, such as Borger et al. (2020) and Xie et al. (2021), used a similar distance for their validation work with respect to ground-based measurements. It has to be noted that our methodology uses only the closest in space satellite

and ground-based observations co-locations for the statistical analysis within the 10 km radius. In practice, this led to a maximum spatial difference between each ground-based station and the respective satellite pixel of up to 5 km.

The next step was to apply the co-location methodology according to which, pairs of co-located satellite and instantaneous ground-based measurements were formed, and their relative percentage difference was calculated as per 100x (TROPOMI-AERONET)/AERONET (%). TROPOMI/S5P passes over most stations once a day, but the ground-based information is instantaneous, meaning that all its observations during each day are available. From the total of available pairs resulting for one day (within a maximum of 10 km in radius), only the one providing the minimum temporal difference, if this temporal difference was up to ±30′, was kept. The criterion of up to ±30′ temporal difference between the satellite and ground-based observation time, is quite strict and is a much smaller time window than what other studies have used (for example, Chan et al. (2020) and Borger et al. (2020) allow up to 2 hours, while Xie et al. (2021) also uses a 30' temporal difference). Its adoption was based on the fact that the AERONET dataset provides clear-sky measurements only, resulting in rather invariable temporally observation fields as far as water vapor is concerned. Still, the number of co-locations resulting from the selection of our criteria is considered adequate to provide solid validation conclusions.

After co-locating the two datasets, a per-station analysis was performed, so as to confirm the choice of the stations to be used for this work depending on the quality and quantity of their data. As an example, the validation results for two individual stations located at different latitudes are shown in Figure 3 and Figure 4. Panels (a) show the monthly mean relative differences between satellite and ground-based observations for the two indicative stations, namely Santa Cruz, Tenerife (Figure 3) and the Acqua Alta Oceanographic Tower (AAOT), Northern Adriatic Sea (Figure 4). The error bars represent the 1σ standard deviation of the means. Panels (b) show the respective scatter plots per station. These two are nice examples of continuous ground-based measurements. It can already be seen from these figures that the Pearson correlation coefficient, R, between ground-based and satellite TCWV observations is above 0.78. The slopes of the linear fits are 0.98 and 1.01 respectively, while the offsets are +0.07 (Figure 3) and -0.02 kg/m$^2$ (Figure 4), respectively. The monthly mean relative biases per station (panels a) for the example stations shown here are within ±0.2 %, demonstrating the good agreement between satellite and ground-based observations, as well as a good temporal stability of both sources of measurement for the available dataset spanning 2.5 years. The variability of the biases, depicted as error bars, may be due to both the ground-based and space-born instrument observational accuracy as well as algorithm and/or meteorology-related effects.

The distribution of the about 70.000 co-locations in space and in time, is shown in Figure 5. The Level 2 data for most stations are uploaded with some delay after observation to the AERONET database, which is the reason for the limited number of available co-locations for the most recent months of the validation period. This is even more pronounced in the Southern Hemisphere, where the number of available stations is smaller, and they extend down to 50° S. There was only one station below that latitude, namely the South Pole Observatory (latitude 90° S), with available measurements that covered only a very short time period of two months during 2018, so it was decided to not be used due to lack of representativeness. Therefore, concerning the Southern Hemisphere we can only draw conclusions for the latitude belt from the equator down to 50° S. The

Northern high latitude co-locations (above 75° N) are available for the summer months of 2018 and 2019 and there are only a few observations for the summer months of 2020.

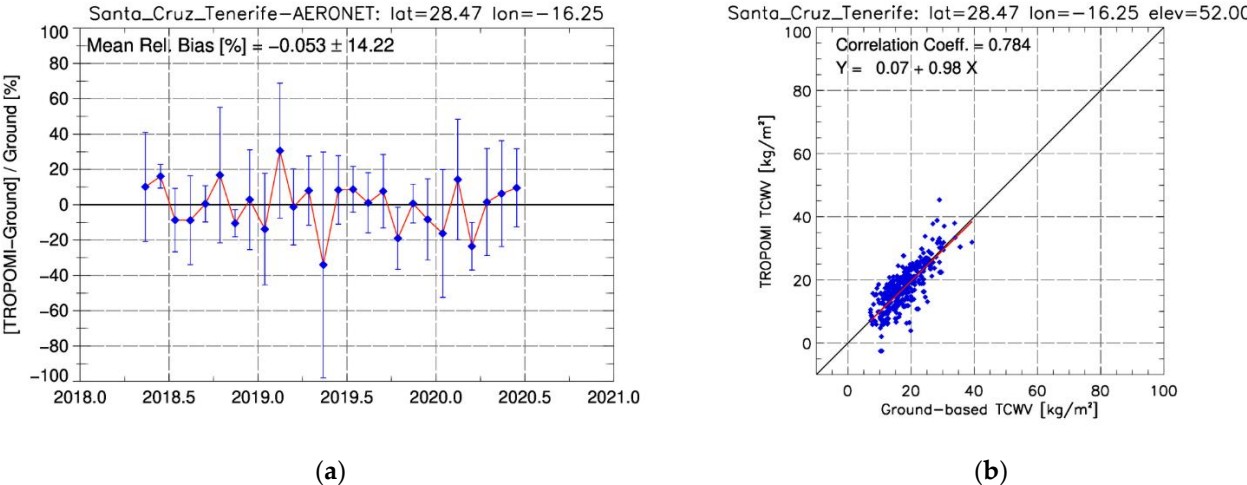

 Figure 3: The monthly mean relative differences between satellite and ground-based observations (panel a) and the respective scatter plot (panel b) for an indicative Northern Hemisphere station, Santa Cruz, Tenerife. The error bars in panel a show the 1σ standard deviations of the means.

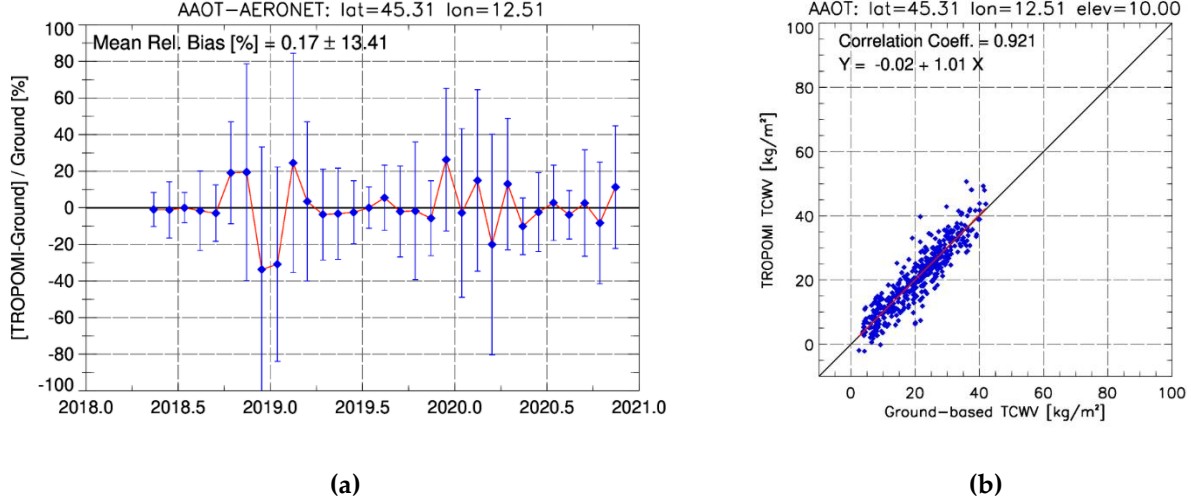

Figure 4: The same as Figure 3, for the Northern Hemisphere tower station AAOT, located at the Northern Adriatic Sea.

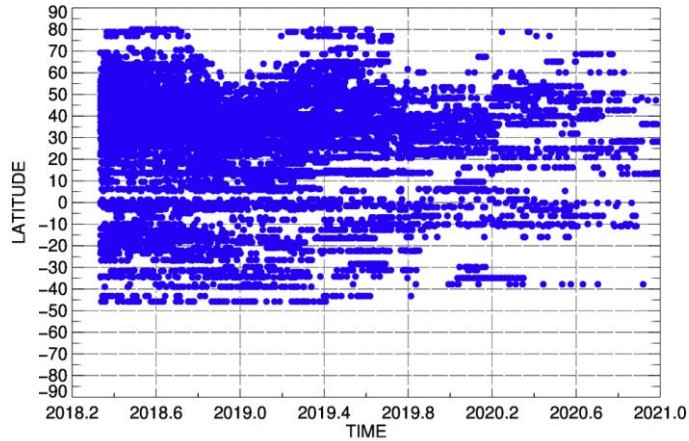

**Figure 5: Spatial and temporal representation of the co-location data used for the validation with ground-based measurements for the time period of the TROPOMI/S5P TCWV data availability (May 2018 to December 2020).**

The monthly means that are shown in the respective time series plots in this work, are calculated by averaging the total number of available instantaneous co-locations per month. The same stands for every averaged parameter plotted here: the mean values are always computed by averaging all individual co-locations that fall within the bin in question. Henceforward, the error bars in the plots (where they are shown) stand for the standard error of the mean with a confidence interval (CI) of 99.7% for each mean value. As expected, since there is a plethora of co-locations, the standard error frequently results to an extremely small

value, showing the good accuracy of the averaging.

## 4. Discussion on the validation analysis

### 4.1. Global comparisons between TROPOMI/S5P TCWV and AERONET ground-based observations

In this section, the archived and quality-controlled AERONET water vapor observations, for the period May 2018 – December 2020, are used for the validation of TROPOMI/S5P TCWV on a global scale. Figure 6 shows the global statistics of the

approximately 70.000 co-located data. The histogram to the left (panel a), shows that the overall mean relative percentage difference between satellite and ground-based measurements is -2.7 %, the 1σ standard deviation is 47.7 % and the standard error is 0.5 %. The distribution of the relative differences around the mean value is a normal Gaussian. Panel b shows the density scatter plot of the co-located datasets. The majority of co-locations have a TCWV content that spans from 0 to 20 kg/m$^2$. The dotted lines show two different approaches for the statistical analysis: the red line is the ordinary least squares

(OLS) method (also used in Figure 3 and Figure 4) and the resulting equation and Pearson correlation coefficient R are shown at the bottom right of the figure; the cyan line represents the total least squares (TLS) method and the respective equation and R are shown at the upper left corner of the plot. Both methods, result to a Pearson correlation coefficient of slightly above 0.9,

which evidences the good overall agreement between the two datasets. The slope of the linear fit is 0.9 (for the OLS; 1.0 for the TLS) and the overall offset between satellite and ground-based observations is +0.9 kg/m$^2$ (for the OLS; -0.6 kg/m$^2$ for the TLS).

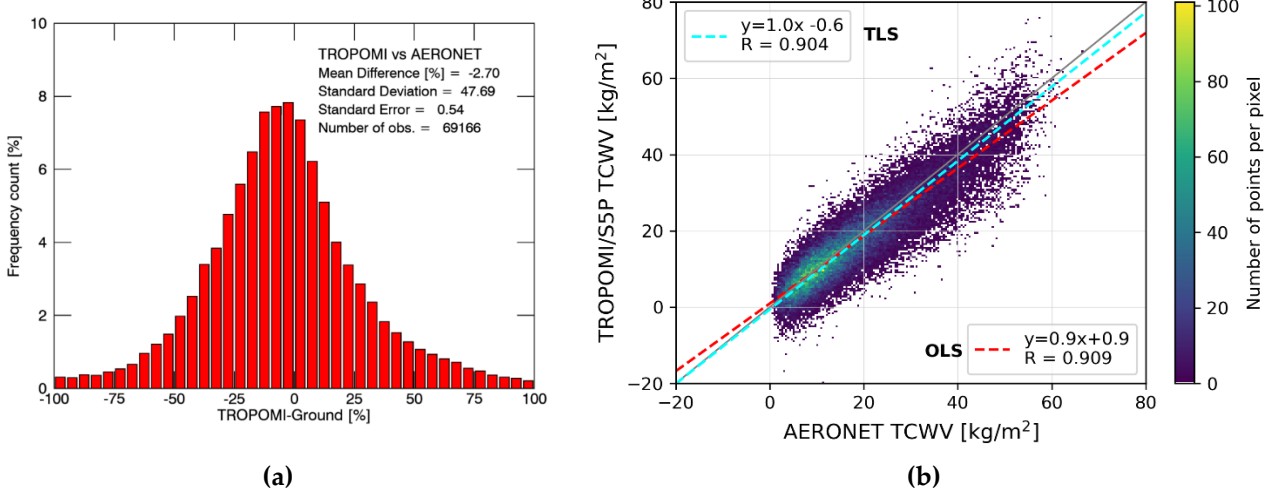

(a)        (b)

**Figure 6: (a) The distribution of the satellite and ground-based co-location relative differences. (b) The density scatter plot showing the correlation between TROPOMI/S5P TCWV and the AERONET observations. The statistical analysis was performed using the ordinary least squares (OLS - red dotted line) and the total least squares (TLS - cyan dotted line) methods.**

To study the temporal evolution of the comparisons, the co-located data are divided into two time-series, depending on the station's latitude. Figure 7 shows the time series of the monthly mean relative differences between satellite and instantaneous co-located (in space and in time) ground-based measurements: panel (a) shows the Northern Hemisphere (NH) time series, while in panel (b) the respective comparisons for the Southern Hemisphere (SH) are depicted, with their standard errors shown as error bars at the 99.7% CI. The illustration of the timeseries in the form of monthly means was adopted because it allows to easily detect any seasonal variability in the comparisons. The NH curve is continuous with no abrupt changes, showing the temporal stability of both sources of measurement, satellite and ground-based, for the 2.5 years of available data. The SH timeseries has a higher variability due to the lower number of co-locations (see Table 1). Since 2020, the number of available AERONET data for this part of the Earth is further reduced, causing the increase of the variability and the standard error of the means. The mean relative bias of the percentage differences for the NH, where the stations density is high, was found to be -3.1 ± 3.2 % and the respective mean standard error of the available monthly means is 3.3 %. In the SH, the mean bias is slightly positive and more variable, +0.9 ± 8.6 %, representing the latitude belt 0° to 50° S. The reduced number of co-locations, results in a higher overall mean standard error of 8.2 %. Table 1 summarizes the global and hemispheric statistics of the monthly mean analysis.

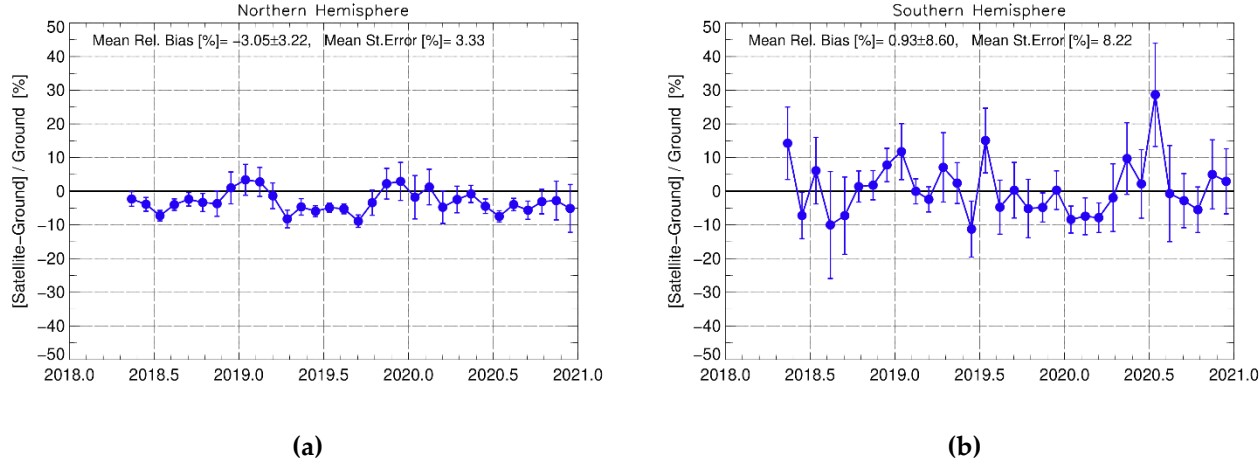

**(a)**                                              **(b)**

**Figure 7: The time series of the monthly mean relative differences between TROPOMI/S5P TCWV and ground-based AERONET measurements, shown for the Northern (panel a) and the Southern Hemisphere (panel b). The error bars stand for the standard error of the mean with a confidence interval (CI) of 99.7%.**

**Table 1: The monthly mean global and hemispheric statistics of the co-located satellite and ground-based observations.**

|                             | NH            | SH            | Globally      |
| --------------------------- | ------------- | ------------- | ------------- |
| Mean Bias ± 1σ              | -3.1 ± 3.2 %  | +0.9 ± 8.6 %  | -2.7 ± 4.9 %  |
| Standard Error (99.7 % CI)  | 3.3 %         | 8.2 %         | 0.5 %         |
| Co-locations                | 58 200        | 11 000        | 69 200        |

The per-station statistical analysis that was performed is shown in the form of a world map in Appendix A, Figure A 1, upper panel, where the mean relative bias in percent for each station is represented by a dot colored depending on the magnitude of its bias. The bottom panels show in greater detail Europe and North America, respectively. It is evident that the vast majority

of the stations have a negative mean relative bias, that reaches -30 % in very few cases, such as L'Aquila, Italy, and Pinehurst, Idaho, USA. On the other hand, there is also a limited number of stations with very high positive biases, up to +30%, like Andenes, Norway and Etna, Italy. Nevertheless, no particular pattern is seen in the mid- and high-latitude stations of either hemisphere. Within the tropics, the mean relative bias per station is mainly negative, ranging between -5 and -25 %. Further statistical analysis on the latitudinal dependence of the mean relative bias between the TROPOMI/S5P TCWV and the

respective ground-based data, is provided in the following paragraphs.

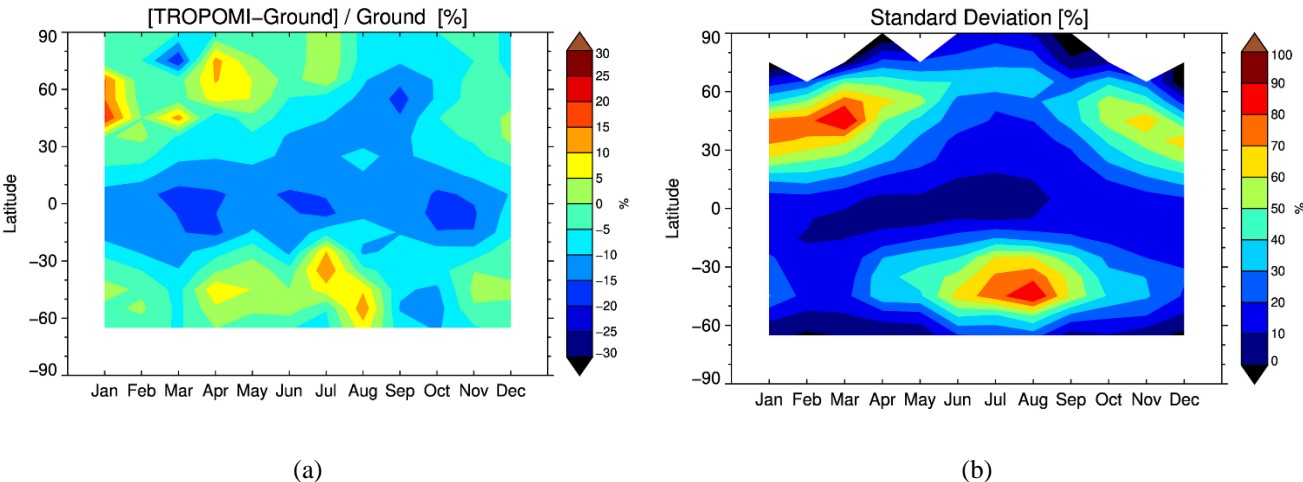

(a)                               (b)

**Figure 8: The seasonal and latitudinal variability of (a) the mean relative differences between satellite and ground-based TCWV observations and (b) the respective standard deviations (in %).**

The contour plots in Figure 8 show the mean relative percentage differences (panel a) and the respective standard deviations (panel b) of the satellite and ground-based co-locations, with respect to latitude and season. Panel a shows that for the mid-latitude winter months of each hemisphere, when the water vapor content of the atmosphere is below ~ 20 kg/m$^2$ (see Figure 1, panel a), the mean relative bias for the respective stations is positive, between 0 and +15 or +20 % (January). During the

310 summer months, when the highest values of water vapor occur for the mid-latitudes, up to 40 kg/m$^2$ (see Figure 1, panel c), the mean relative bias is within ± 5 %. In the tropics, where the TCWV content is higher throughout the year (40 – 80 kg/m$^2$), the mean relative bias is constantly negative, ranging between -5 and -20 %. Panel b depicts the strong seasonality of the comparisons' standard deviations, i.e. of their variability, which is high during the mid-latitudes winter months of both hemispheres (up to ~ 90 %), and lower (10-30 %) during summer, when the number of ground-based AERONET

measurements (i.e. the number of co-locations) and their accuracy is much higher (Fragkos et al., 2019). Additionally, as shown in Figure 1, for latitudes higher than the tropics of both hemispheres the water vapor content is lower and has a stronger temporal variability, explaining the higher standard deviations of the relative differences. It is also interesting to see that the variability of the comparisons for the tropics (Fig. 8, 15° N-15° S) is much lower (up to 20 %) compared to the other latitude belts, showing that the negative mean relative bias of our comparisons is temporally invariable in this part of the globe, where

the water vapor content is higher than ~ 40 kg/m$^2$.

To further investigate the latitudinal patterns of our comparisons, the mean relative percentage differences per station with available ground-based data are averaged in 10° latitude belts and are shown versus latitude in Figure 9, panel (a). The same is also shown in panel (b), but the averaged parameter per latitude bin is the difference between satellite and ground-based observations in kg/m$^2$. The overall mean relative percentage bias for the latitudinal dependency is -1.1 ± 6.1 % and has a mean

standard error of 6.8 %. The agreement between satellite and ground-based observations remains within ± 10 % for individual belts of the NH and the belt northwards 50° of the SH. The latitude bins -30° S to 30° N form a U-shaped curve, showing that the satellite instrument reports lower TCWV up to ~10 % with respect to ground-based observations close to the equator and reaching ~0 % at ±30°. This result, which corresponds to a difference between satellite and ground-based observations up to - 4 kg/m$^2$ (panel b), is in agreement with Chan et al. (2022), where the dry bias is attributed to albedo effects in the visible band over vegetation and to the presence of aerosol and/or clouds in the measurement field. For the NH high latitude stations, above 70° N, the discrepancy becomes positive up to 10 % and has a very large standard error due to the limited number of co-locations (panel a). In terms of difference (panel b), this percentage accounts for a small overestimation of less than 1 kg/m$^2$ by TROPOMI/S5P occurring close to the poles where the amount of water vapor is less than 20 kg/m$^2$ (see Figure 1). Considering that the uncertainties of both types of measurement is ~10 %, the comparison of the satellite and ground-based observations is regarded as satisfactory. The performance of the TROPOMI/S5P TCWV retrieval algorithm with respect to the surface albedo parameter which significantly changes with latitude is currently adequate but could be further improved in the future, as is also shown further on in this work (Fig. 12, panel b).

The statistics per latitude belt in terms of mean difference (satellite – ground) in kg/m$^2$, mean relative bias ± standard deviation and mean standard error of the comparisons (in %), are shown in Table 2 for 15° latitude belts up to ± 30° N and S. The belts 0 to 15° N and 0 to 15° S represent the tropics. Above 30°, the binning is doubled because due to the low water vapor content and its low variability, the differences in the statistics between belts 60°N-75°N and 75°-90°N would be negligible.

It is worth noting that the mean relative bias of each latitude bin and the respective mean standard deviation, thus the variability (not shown in Figure 9), should not be attributed to the satellite product only, since it is well known that some ground-based stations may overestimate or underestimate their observational constituent systematically due to the meteorological conditions occurring at the station site. Moreover, when such a station does not provide a continuous record of observations, there is a high possibility that it will introduce an artificial and non-representative bias to the validation. Most of these stations, that did not fully cover the time period of our study, were filtered out of the ground-truth database used in this work. Nevertheless, for some latitude bins, like 40° S to 50° S, where the station density or the temporal coverage is low, the respective stations were considered with the remark that the statistics resulting from their co-locations should be interpreted with caution.

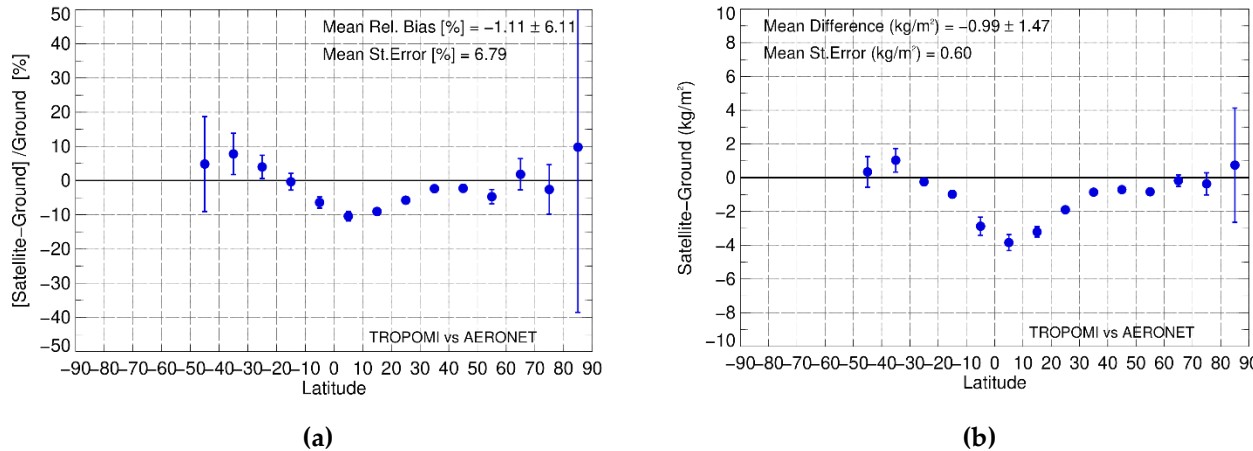

**Figure 9: Panel (a): The relative percentage differences between co-located TROPOMI/S5P TCWV measurements and ground-based observations from AERONET instruments plotted versus latitude. Panel (b): As in panel (a) but for the differences between satellite and ground-based observations in kg/m². In both panels the error bars show the standard error of the mean with a confidence interval (CI) of 99.7%.**

**Table 2: The zonal statistics of the co-located satellite and ground-based observations**

| Hemisphere | Latitude belt | Mean Diff.[1] (kg/m²) | Mean St. Err.[2] (kg/m²) | Mean Rel. Bias (%) | Mean St. Err.[2] (%) |
|---|---|---|---|---|---|
| NH | 90°-60° | -0.0 ± 1.2 | 2.0 | -6.2 ± 29.9 | 37.41 |
| | 60°-30° | -0.6 ± 0.7 | 0.4 | -1.8 ± 4.6 | 4.2 |
| | 30°-15° | -2.4 ± 1.1 | 1.2 | -6.8 ± 3.5 | 4.9 |
| | 15°-0° | -3.5 ± 1.5 | 2.1 | -9.6 ± 4.2 | 5.8 |
| SH | 0°-15° | -2.2 ± 1.4 | 2.1 | -5.0 ± 9.5 | 10.31 |
| | 15°-30° | -0.6 ± 1.2 | 1.0 | -2.8 ± 12.1 | 11.4 |
| | 30°-50° | +0.6 ± 2.0 | 3.3 | +4.8 ± 18.8 | 30.4 |

[1] Satellite-Ground

[2] 99.7% CI

**4.2 Discussion on the dependence of TROPOMI/S5P TCWV on various geophysical parameters**

In this section, the dependence of the validation results on different variables is investigated. These quantities can be parameters that are used as inputs for the TCWV retrieval algorithm, such as cloud and surface information, or algorithm-related parameters, like the air mass factor. To inspect any possible dependences, all available co-locations are averaged in bins regarding the parameter in question. Note that, in the following figures, when the number of co-locations that are averaged for

each bin is less than 3 % of the total, the respective data point is shown in gray (instead of blue). This is a way to distinguish the data points in terms of relative importance.

### 4.2.1. Viewing geometry dependency

Figure 10 shows the dependency of the relative differences on solar zenith angle (SZA – panel a) as provided by TROPOMI/S5P and the satellite pixel number (panel b). Regarding the dependency on SZA (panel a), TROPOMI/S5P reports
lower TCWV than the AERONET observations by up to -7 % for SZAs below 45°, where more than half of the co-locations are contained. Their difference is eliminated for SZAs 45°-70° and slightly increases with SZA, reaching +9 % above 70° for a limited number of co-locations. Overall, the dependence of the relative differences on SZA is ~16 % peak-to-peak, or ~ 7 % if only SZAs below 70° are considered. As expected, the standard error of the means increases for larger SZAs because of the increase in the uncertainty of the measurements, the lower number of co-locations and of course due to the strong effect of the
winter mid-latitude co-locations.

As for the dependence of the comparisons on satellite pixels (panel b), pixels 0-10 have a positive mean relative bias of +7 % and pixels 400 – 450 have a systematic negative mean relative bias of 8.2 %. Except for these two areas of pixels, the illustration of the dependence shows a variability within 0 and -10 %, so there is no evident overall systematic east-west dependence in the TROPOMI/S5P swath.

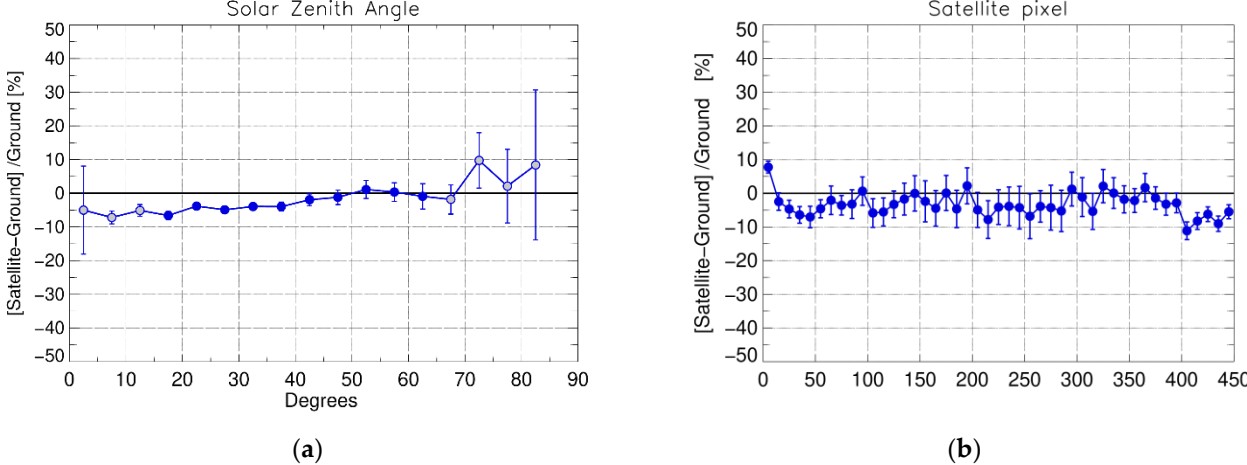

(a)  (b)

**Figure 10: The dependence of the mean relative differences between satellite and ground-based TCWV observations on solar zenith angle (panel a) and the satellite pixel number (panel b). In both panels the error bars show the standard error of the mean with a confidence interval (CI) of 99.7%. Grey dots represent bins containing < 3% of the total co-location pairs.**

## 4.2.2. Input data dependency

The TCWV retrieval algorithm requires two categories of input data that are simultaneously retrieved from TROPOMI/S5P measurements: the cloud properties and the surface properties.

Concerning the cloud properties, retrieved with the OCRA/ROCINN algorithms (Loyola et al., 2018), Figure 11 shows the dependence of the comparisons on cloud top pressure (panel a), cloud albedo (panel b) and cloud fraction (panel c). As is indicated from the figures, the satellite TCWV has a noticeable dependence on cloud pressure and cloud albedo:

- For cloud top pressures (panel a) up to 800 hPa, the data bins with relatively high number of co-locations have a positive bias of ~ +5 to +10 %, which decreases to -20 % when the pressure increases to ~900 or 1000 hPa, hence for clouds of lower height, that may also affect the ground-based measurements. Borger et al. (2020), that validated their TCWV product against Special Microwave Imager Sounder (SMISS) on board f16 and f17, ERA-5 and GPS data, examined the dependence of their comparisons on cloud height. They also found that low clouds, located below 3-4 km, cause an underestimation in the retrieved TCWV of about -13 %. Typically, the cloud top pressure of 800 hPa that we found to be the turning point, corresponds to ~ 2-3 km, therefore our results are very consistent to Borger et al. (2020), especially considering the fact that they are based on a different retrieval algorithm.

- Panel b shows that the comparisons have a strong overall dependence of 40 % peak-to-peak on cloud albedo, i.e. on the fraction of solar radiation that is reflected directly by clouds in the atmosphere (Glossary of Meteorology, AMS, 2022). The bias of the comparisons is positive, +15 % maximum, for cloud albedo values below 0.3 and becomes negative, up to -25 %, for increasing cloud albedo, thus for brighter clouds.

- The cloud fraction figure (panel c) shows that the vast majority of the co-locations have cloud fraction values below 0.3, which is expected since both satellite and ground-based observations are filtered for cloudiness (satellite data are filtered for cloud fraction < 0.5). Within the cloud fraction range of 0 to 0.3, no particular dependence is seen. The co-locations that are characterized with cloud fraction values between 0.3 and 0.5 are very few in population but they introduce high positive mean relative biases. Filtering-out the co-locations dataset for cloud fraction over 0.3 was also investigated, resulting to no major differences in the overall validation results due to the limited contribution of the relatively low number of co-locations with cloud fraction values in the range 0.3 – 0.5. Nevertheless, it is advisable to not use this small portion of TCWV data for future scientific studies.

Overall, the dependence of the relative percentage differences on cloud fraction and cloud albedo could also be an issue of the ghost total column, i.e., the water vapor that may be present beneath the clouds but not properly measured by the satellite instrument when even a part of the sky is cloudy. The fact that the ground-based measurements are screened for cloudiness and the satellite observations are allowed to have a part of the measurement field covered with clouds, can be another cause for the differences found between them.

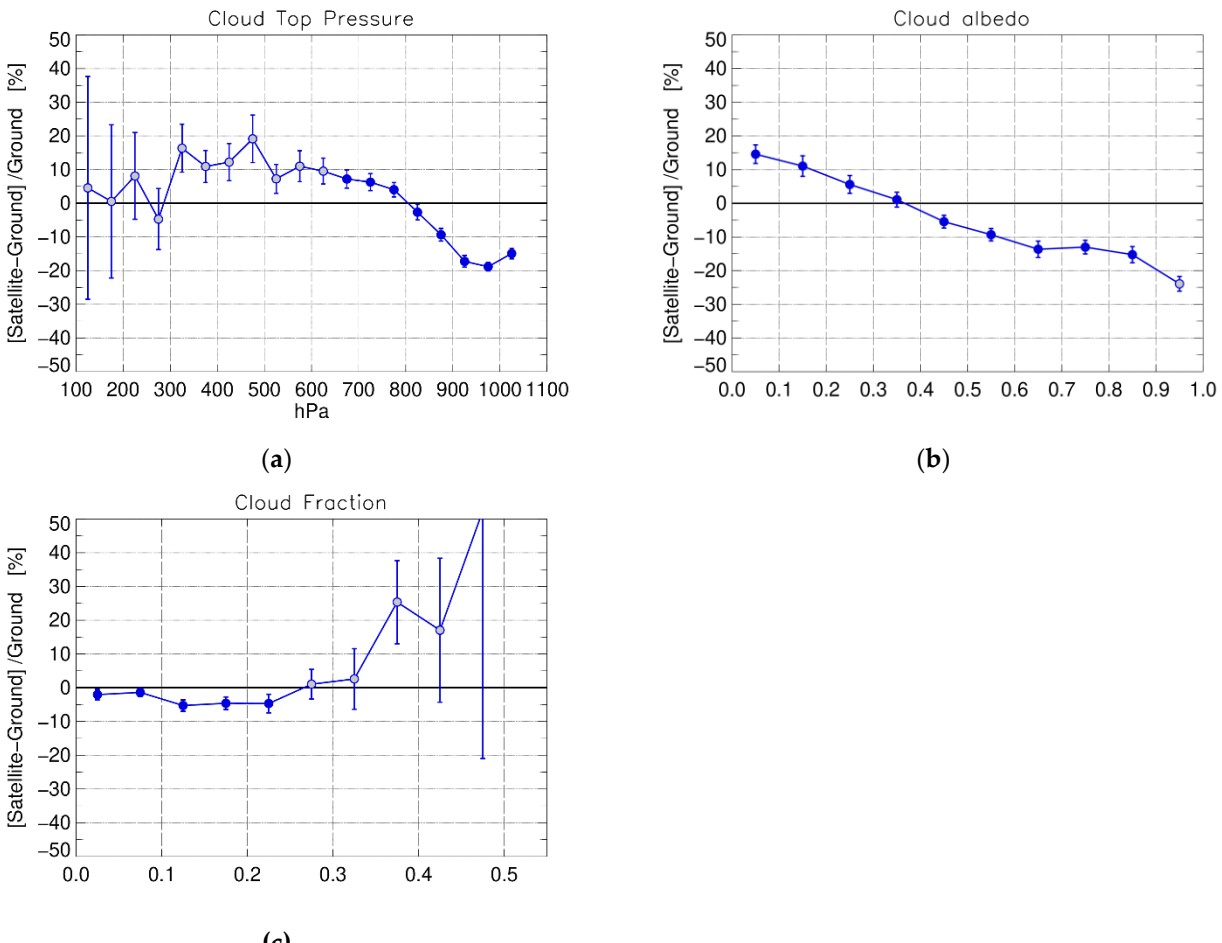

**Figure 11: The dependency of the comparisons of satellite to ground-based TCWV measurements on three different cloud parameters, namely: (panel a) cloud top pressure; (panel b) cloud albedo and (panel c) cloud fraction. The error bars show the standard error of the mean with a confidence interval (CI) of 99.7%. Grey dots represent bins containing < 3% of the total co-location pairs.**

The surface properties used for the TROPOMI/TCWV product, namely surface pressure and surface albedo, were retrieved with the GE_LER algorithm (Loyola et al., 2020). The dependence of the comparisons on these properties is shown in Figure 12:

- Surface pressure (panel a): for the typical range of surface pressures i.e., 900 – 1050 hPa, no systematic dependence is seen in the comparisons. As expected, the bins with pressures less than 900 hPa have a limited number of co-locations and the curve represents mostly noise data.

- Surface albedo (panel b): As the density of ground-based stations is much higher at the mid-latitudes of both hemispheres, very few co-locations have surface albedo above 0.2 and since they showed no apparent systematic

dependence on surface albedo, they are not included in the figure. For surface albedo values below 0.2, the relative

differences range within ±10 %, but no systematic dependence is detected.

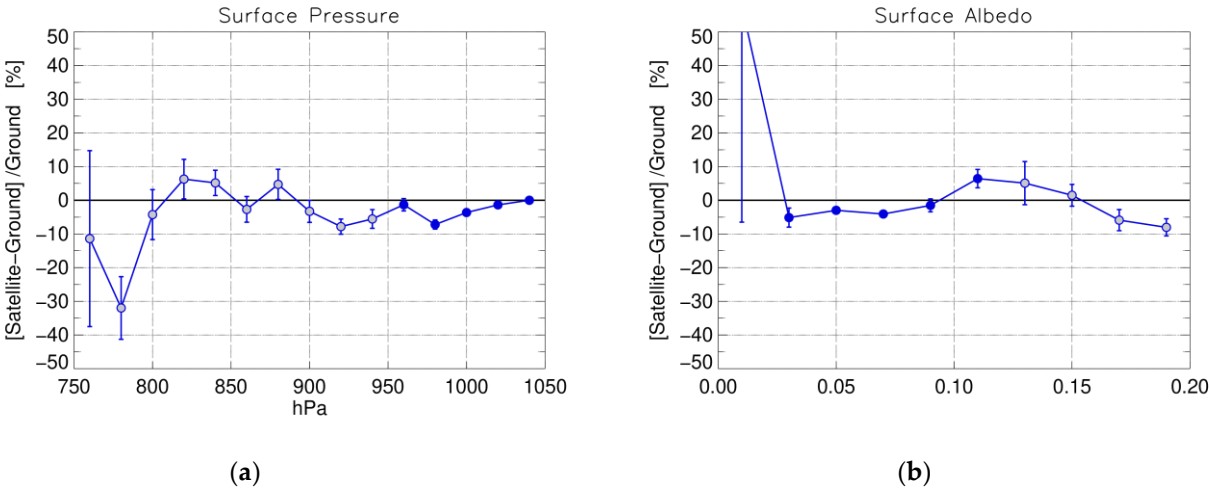

(a)             (b)

**Figure 12: The dependence of the comparisons of satellite to ground-based TCWV measurements on two surface parameters, namely: (panel a) surface pressure and (panel b) surface albedo. In both panels the error bars show the standard error of the mean**
**with a confidence interval (CI) of 99.7%. Grey dots represent bins containing < 3% of the total co-location pairs.**

### 4.2.3. Dependency on algorithm-related parameters

The following parameters are related to the retrieval algorithm of the TROPOM/S5P TCWV data:

- Air Mass Factor (AMF). The dependence of the relative differences of co-located data on AMF is shown in Figure
13, panel a. For the well-populated bins with AMF ranging between 1 and 2 the bias is negligible, up to -5 %, which
  is expected since the measurements acquired under low SZAs have also a bias of 0 to -5.5 % (Figure 10, panel a). For
  AMF values between 2 and 3 the bias becomes negative, up to -20 %, probably affected by cloudiness, while for
  AMFs greater than 3 the number of co-locations per averaged bin is very low and their variability is not considered
  statistically important.
- Root Mean Square error of fit (RMS). Figure 13, panel b, shows no systematic dependence on RMS, even for the bins
  with a low number of co-locations.
- Water Vapor Slant Column Density (SCD). Figure 13, panel c, shows the dependence on the Water Vapor SCD. No
  dependence of the comparisons on the specific parameter is seen when the algorithm retrieves values above 10 kg/m$^2$.
  The limited number of co-locations with positive SCD values below 10 kg/m$^2$ have a relative percentage difference
of -32 %. Negative SCDs are mainly due to measurement noise. When the spectral fit residual is analyzed, the random
  uncertainty of SCD retrieval is typically less than 1 kg/m$^2$. Although this error is small, it could cause a significant

impact over areas with low atmospheric water vapor content and result in negative values. In addition, this effect can be further amplified when the AMF is small (below 1). Even though very few retrievals are based on an AMF value of less than 1, we note that comparisons to these pixels result in extremely high relative differences with respect to ground-based measurements, up to 100 %.

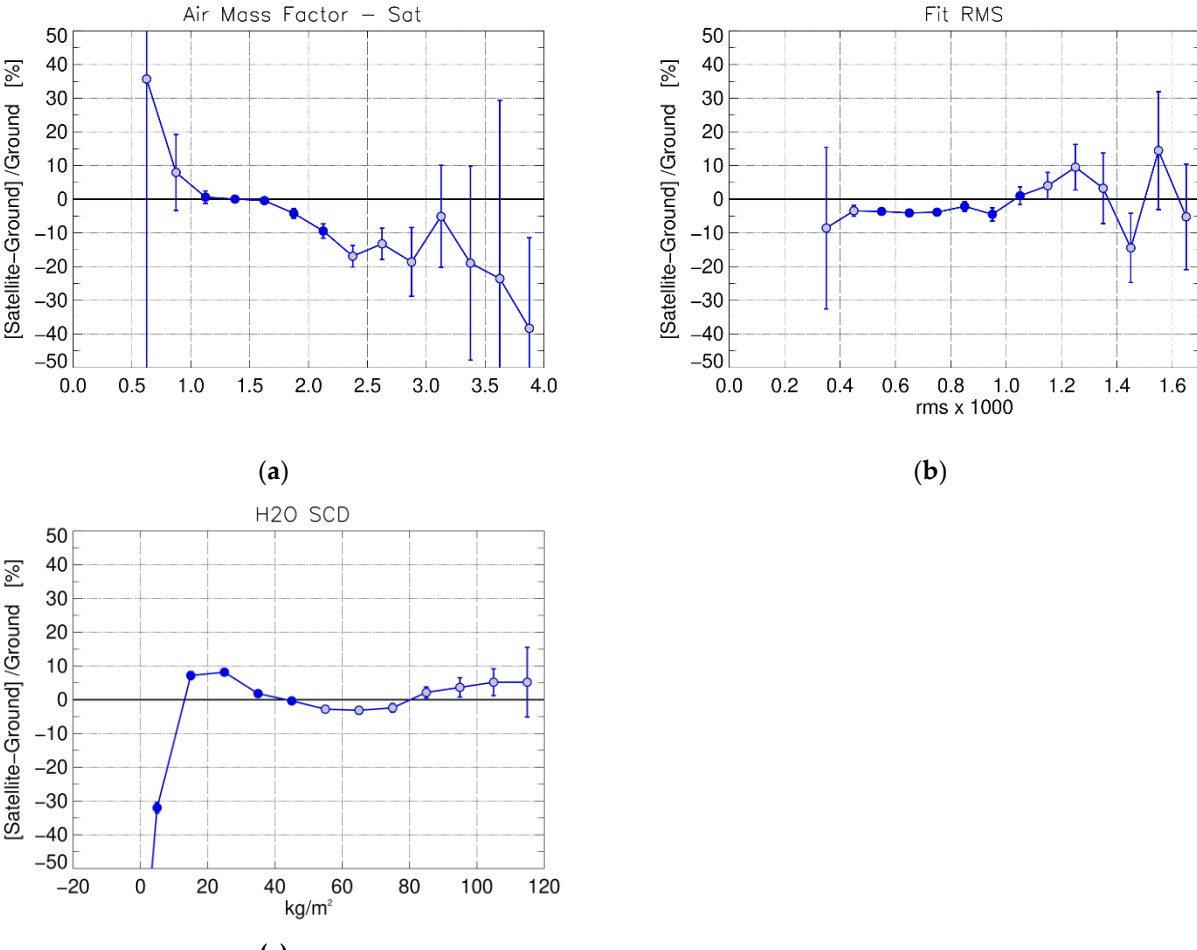

(a)

(b)

(c)

Figure 13: The dependence of the comparisons of satellite to ground-based TCWV measurements on three parameters, namely: TROPOMI/S5P (a) air mass factor; (b) RMS and (c) water vapor slant column density. Grey dots represent bins containing < 3% of the total co-location pairs.

## 5. Summary and conclusions

The main purpose of this work is to examine the performance of the new TCWV product retrieved from the blue band of the TROPOMI/S5P observations and their consistency with AERONET ground-based measurements. About 70.000 instantaneous co-located data were available during the time period May 2018 to December 2020, originating from 369 ground-based stations

and the respective satellite overpasses. The relative percentage differences that were calculated from the co-located pairs of data with temporal difference up to ±30' and maximum search radius up to 10 km, correspond to clear-sky observations (satellite cloud fraction <0.5) and were statistically analyzed in terms of temporal and latitudinal dependences. Furthermore, their dependence on various parameters was investigated. The validation results can be summarized as follows:

- The overall mean relative difference between TROPOMI/S5P and AERONET TCWV observations is -2.7 %, while the Pearson correlation coefficient is 0.91. When the two hemispheres are studied separately, their mean bias results to -3.1 % for the NH and +0.9 % for the SH. Considering that the uncertainty of the satellite TCWV product is ~ 10-19 % (tropics), and the ground-based measurements' uncertainty is reported to be ~10 %, the agreement between the two datasets is deemed very satisfactory. The mean standard error of the comparisons at a 99.7% CI is 0.5 %, highlighting the good consistency of the results. Additionally, considering the dry bias of the AERONET observations that was discussed in Sect. 2.2, which is about -5 to -10 % (depending on the study and its reference) and varies with season and latitude, it can be concluded that the satellite TCWV observations have a dry bias with respect to the "absolute" truth of about -8 to -13 %, respectively.

- A zonal analysis of the monthly mean relative differences and their standard deviations showed that for mid-latitudes, where water vapor is low ($10 - 20$ kg/m$^2$) and more variable with season, TROPOMI/S5P overestimates TCWV by ~ 5-10 % (i.e. $1 - 2$ kg/m$^2$) during winter months of each hemisphere. The variability (standard deviation) of these overestimations is very high, up to 90%. During summer months, when the water vapor content over mid-latitude stations is higher (up to ~ 40 kg/m$^2$), the availability of the ground-based measurements is increased and their uncertainty is lower, the relative differences between TROPOMI/S5P and AERONET are within ± 5 % (i.e. up to ± 2 kg/m$^2$) and have a rather limited variability (~10-30 %). In the tropics (15° N-15° S), where the bulk of the water vapor is concentrated and is quite stable annually, our analysis confirmed that TROPOMI/S5P has a dry bias of up -10 % (or up to -4 kg/m$^2$) with respect to the ground-based measurements (see Table 2), that is temporally stable with a variability of less than 20 % (Figure 8, panel b).

- Finally, many parameters influencing the satellite retrievals were studied, and no particular dependences were found, except for a dependency on cloud top pressure (CTP) and cloud albedo. Specifically, it was shown that for low cloud top height (CTP > 800 hPa) the satellite reports lower TCWV by up to -20 % compared to the ground-based measurements. The dependency on cloud albedo is also strong, about 40 % peak-to-peak, showing a wet bias of 15 % when the cloud albedo is below 0.3 and a dry bias up to -25 % when the clouds are more reflective (albedo > 0.3).

To conclude, as shown from the validation of 2.5 years of available satellite observations, with respect to ground-based observations from AERONET, the TROPOMI/S5P TCWV product retrieved from the blue spectral range, is a temporally stable product of high quality and precision, especially at the tropics. Also, it is not significantly affected by any other parameters, except from clouds when and if some cloudiness at lower atmospheric layers is present in the measurement field. This product is expected to substantially contribute to a long time series of total column water vapor climate data record achieved by utilizing other blue band satellites, such as GOME (Global Ozone Monitoring Experience; Burrows et al., 1999),

GOME-2 (Global Ozone Monitoring Experience 2; Callies et al., 2000), SCIAMACHY (SCanning Imaging Absorption SpectroMeter for Atmospheric CHartographY; Bovensmann et al., 1999), and OMI (Ozone Monitoring Instrument; Levelt et al., 2006), along with TROPOMI/S5P.

**Appendix A**

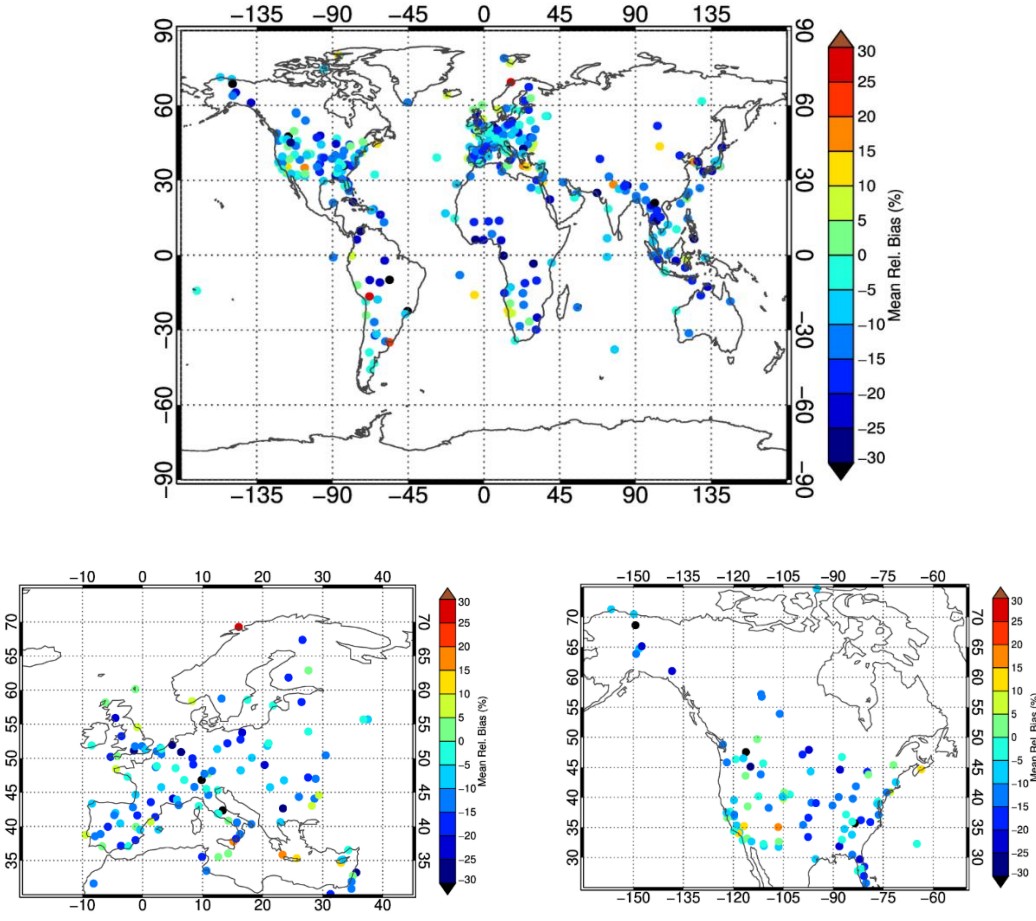

**Figure A 1: The mean relative bias (in %) between satellite and ground-based observations per station in the form of a world map (above). The two panels below show the very-well station populated areas of Europe (left panel) and North America (right panel) in greater detail.**

**Data availability:** The TROPOMI/S5P TCWV data were retrieved using the algorithm described in Chan et al. (2022). The dataset is foreseen to be available through the ESA Sentinel 5 Precursor Product Algorithm Laboratory (S5P-PAL) framework.

The AERONET ground-based Level 2.0 precipitable water measurements were downloaded from https://aeronet.gsfc.nasa.gov/.

**Author contribution:** Conceptualization by KG and DB. The validation methodology was defined by KG and MEK. The scripts used for the analysis were written by KG and MEK. The data analysis and validation were performed by KG, MEK and DB. KLC and DL provided the satellite data. KG wrote the manuscript. MEK, DB, DL and KLC reviewed and edited the manuscript. Project administration by DL. Funding acquisition by DL and DB. All authors have read and agreed to the published version of the manuscript.

**Competing interests:** At least one of the (co-)authors is a member of the editorial board of Atmospheric Measurement Techniques. The peer-review process was guided by an independent editor, and the authors have also no other competing interests to declare.

**Acknowledgements:** We thank the AERONET PI(s) and Co-I(s) and their staff for establishing and maintaining the 369 sites used in this investigation. We thank the European Space Agency (ESA) for supporting the TROPOMI/S5P TCWV optimization within the framework of Sentinel 5 Precursor Product Algorithm Laboratory (S5P-PAL) project.

**Financial Support:** This research was funded by the German Aerospace Center (DLR) in coordination with the DLR Innovative Products for Analyses of Atmospheric Composition (INPULS) project.

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
