# Peer review of "TROPOMI/S5P Total Column Water Vapor Validation against AERONET ground-based measurements"

_Atmospheric Measurement Techniques, 2022_

## Author Comment (AC1)

**Review: "TROPOMI/S5P Total Column Water Vapor Validation against AERONET ground-based measurements" by Garane et al.**

**General comments**

In this study, Garane et al. validate a new TROPOMI TCWV product by Chan et al. (2022) using 2.5 years of ground-based measurements from AERONET as reference. They also investigate the influence of different input variables (e.g., viewing geometry, surface albedo, clouds) and the retrieval results themselves (i.e., $H_2O$ SCD and AMF) on the retrieval performance.

Although the overall aim of the paper is interesting, I have some concerns about the validation analysis. I also feel that the authors miss an opportunity by not taking advantage of the high density of AERONET stations in certain regions and instead averaging the results zonally, which unfortunately also results in a lot of information being lost.

That being said, I recommend publication if the following points and concerns are addressed.

Thank you for recommending our paper for publication.
* * *
**Major issues**

1. It is not completely clear how the collocation of ground-based and satellite measurement was conducted. If I look at the total number of data and roughly calculate, I results in about 2 measurements per day per station (633000 / (365*351*2.5) ~ 2). And here all filters are already taken into account (CF<0.5, AMF, RMS, SZA). In my opinion, this number seems unexpectedly high, especially considering that not all stations were able to provide measurements for the most recent months. The authors must explain the collocation procedure more clearly: were several satellite pixels (within 10km to the reference station) compared with one measurement from AERONET? Or is simply a large fraction of the collocated measurements at stations at high latitudes? If the former is the case, then some reference measurements would be used more often than others, making the comparison inconsistent. I would therefore suggest either to take only the closest satellite pixel within 10km or to calculate the mean value of all satellite pixels within 10km and compare it with AERONET.

   **Reply:** We thank the reviewer for pointing out this issue. When the first validation exercises were performed, based on a limited dataset that was available at the time, we used the following co-location technique to increase the sample of the data to be evaluated:

Each satellite measurement from a specific pixel (the one spatially closest to the ground-based station within 10km), was compared to all instantaneous ground-based measurements that were performed within a ±30′ temporal interval with respect to the time of satellite observation. This way, for each overpass instance there were more than one matching pairs, explaining the high number of co-locations.

This approach is not necessary now that a significantly larger satellite dataset of 2.5 years, is available. Therefore, the co-location methodology was changed to keeping only the match with the minimum temporal difference between satellite and ground-based observations within a 10 km radius, if this temporal difference is up to 30 min. The resulting total number of co-locations is now about 70.000. The new methodology was applied throughout the manuscript and all plots and statistics were updated and the manuscript was revised accordingly. Additionally, a paragraph with a detailed description of the co-location methodology is added in Section 3.

Furthermore, negative TROPOMI TCWV values also appear in the comparisons (for instance in Figure 3b and 6b). I would ask the authors to clarify where these negative values come from.

**Reply:** It is very common that satellite atmospheric observational data occasionally show some negative values. In this case, it happens when the water vapor content is very low (close to zero), and due to the instrument noise, the spectral fit might retrieve slightly negative SCD. These negative values are well within their measurement uncertainties. In addition, it is not a good idea to simply ignore all the negative values. As the measurement uncertainty is supposed to follow the normal distribution (assuming a random instrument noise), if we ignore all the negative values, it will result a positive bias in the averages.

We have also revised Figure 6b to better indicate the fraction of negative values in the scatter plot (see Figure 1, in this document).

2. The use of zonal means does not make much sense, as TCWV has a high variability along a longitude. This also negates the great advantage of AERONET, namely the network's high station density. I suggest the authors to restructure Section 4 as follows:

- One should carry out the analysis of TROPOMI vs. AERONET (i.e. the regression analysis) for each station individually and then present the fit results on a world map.

- For regions with a high station density (e.g. Europe and North America), separate plots could be shown separately.
- In the corresponding regions, one could interpolate the regression results of all stations and then analyse how well the performance of the retrieval depends on geophysical parameters, i.e. performance in humid/arid areas, influence of albedo, ground elevation, etc. This would also lead to an overall better understanding of the retrieval.

> **Reply:** Thank you for the comment and your suggestions.
>
> As suggested, the three maps (world, Europe and N. America) were added in Appendix A and commented in the manuscript: the latitudinal and longitudinal variability of the mean relative bias was presented in the form of a world map showing the relative mean bias of each station using a colorbar. To further investigate any possible patterns in relative bias over Europe and N. America, where the stations are very dense, the two areas were also plotted separately. As it was stated in the text:
>
> *"… no particular pattern is seen in the mid- and high-latitude stations of both hemispheres. Within the tropics, the mean relative bias per station is mainly negative, ranging between -5 and -25 %."*
>
> Additionally, a panel with a new contour plot was added in Figure 8 (panel a), showing the seasonal and latitudinal variability of the mean bias, as it was suggested by Reviewer #2.

3. It is not really clear why AERONET is used at all when the authors themselves say that AERONET most likely underestimates the actual TCWV content (see Section 2.2) and the quality of the measurements also strongly depends on the calibration of the instrument on site (see line 166f). What is the great benefit of AERONET compared to other measurement networks? GPS measurements from SuomiNet or IGS, for example, have a much higher accuracy than AERONET and can also measure under all-sky conditions.

> **Reply:** We thank the reviewer for this comment.
>
> The main reason for basing our work on AERONET water content observations is that, first of all, the network is very well established, with more than 25 years of operations and a transparent data quality assurance plan through its extensive cal/val routine operations, see here: System Description - Aerosol Robotic Network (AERONET) Homepage (nasa.gov). Furthermore, the network offers a complete global coverage,

covering the entire planet quite satisfactorily, see here: AERONET Data Display Interface - WWW DEMONSTRAT (nasa.gov) .

Even though it is true that AERONET has been extensively used for AOD validation, the AERONET water vapor observations have also been employed in space-born and ground-based instrumentation validation studies, see for e.g. https://doi.org/10.1016/j.atmosres.2019.04.005, https://doi.org/10.3390/rs13163246, https://doi.org/10.5194/amt-11-81-2018, etc. Due to the fact that a number of studies have already utilized the ground-based GPS datasets, and the same TCWV from TROPOMI/S5P product was also validated against GNSS (https://doi.org/10.3390/atmos13071079), in this work we aimed to investigate the potential provided by the AERONET TCWV. Your suggestion is of course very welcome for future works.

As for the calibration issues mentioned in the first version of the manuscript, it must be noted that the wording used in the text does not reflect what was actually meant. Of course, using Level 2.0 excludes any calibration issues from the discussion. The sentence was rephrased as follows:

*"The monthly mean relative bias per station (panels a) depends strongly on the ground-based instrument's operation and maintenance, …"*
* * *
**Other**

Overall, the quality of the figures needs to be significantly improved: Instead of point clouds, 2D histograms should be used (e.g. Figure 5 or 6b). The numbers in Figures 10-13 are hardly readable. It might be better to show the amount of data points using a colorbar with coloured dots.

**Reply:**

- With respect to Figure 5, which demonstrates not only the availability of data per latitude, but also the availability as time progresses, we are unable to consider a different representation, one that will keep both pieces of information. The quality of the figure was improved.
- Figure 6b was changed to a density scatter plot of a much better quality (see Fig.1 of this document).

- Regarding Figures 10-13, and the illustration of the data points with a very low number of co-locations, they were modified and we used the following sentences in Section 4.2, 1st paragraph, to clarify the new way of depiction:

  *"Note that, in the following figures, when the number of co-locations that are averaged for each bin is less than 3% of the total, the respective the data point is shown in gray (instead of blue). This is a way to distinguish the data points in terms of relative importance."*

Moreover, the language should be improved. Here and there the wording is not really appropriate. For example, "quantity" should be replaced by "variable" and "percentage difference" by "relative difference" throughout the paper.

**Reply:** Thank you for your comment. The wording was changed according to the suggestions.
* * *
**Specific comments**

L11: "blue wavelength band" to "visible blue spectral range"

**Reply:** The phrasing was corrected.

L12: MetOp

**Reply:** The wording was changed in lines 11 and 12.

L18: -3%: Table 2 shows much higher values (-4 to -10% in NH, +2 to +6% in SH).

**Reply:** This percentage referred to the mean relative bias for the mid-latitudes and the tropics and resulted from averaging the mean relative biases of all latitude belts within ±60°.

Table 2: The zonal statistics of the co-located satellite and ground-based observations

| Hemisphere | Latitude belt | Mean Diff.[1] (kg/m$^2$) | Mean Rel. Bias (%) | Mean St. Dev. (%) | Mean St. Err.[2] (%) |
|---|---|---|---|---|---|
| NH | 90°-60° | -0.4 | 1.2 ± 31.5 | 61.3 | 12.6 |
|  | 60°-30° | -0.8 | -4.0 ± 2.9 | 44.0 | 1.3 |
|  | 30°-15° | -2.2 | -5.9 ± 3.4 | 23.6 | 1.6 |
|  | 15°-0° | -3.7 | -9.6 ± 3.0 | 18.5 | 2.0 |
| SH | 0°-15° | -2.5 | -5.9 ± 5.5 | 32.2 | 3.3 |
|  | 15°-30° | -0.7 | 2.4 ± 8.3 | 52.3 | 3.6 |
|  | 30°-60° | +0.5 | 5.8 ± 12.3 | 46.1 | 8.9 |
|  | 60°-90° | +0.3 | 42.2 ± 4.9 | 84.8 | 16.5 |

[1] Satellite-Ground

[2] 99.7% CI

The abstract was revised and now the sentence refers to the overall mean relative bias, which is -2.7 %, after the new analysis that followed the revision of the co-location methodology:

*"The Pearson correlation coefficient of the two products is found to be 0.91 and the mean bias of the overall relative percentage differences is of the order of only -2.7±4.9 %."*
Please also note that the statistics in Table 2 were updated.

L21: "low cloudiness" --> low cloud heights
**Reply:** The wording was changed.

L23: "-4 +- 4.3 % with the ground-truth": In Section 5 it is written that it is -9 to -13%. Accordingly, one should write here that it is -4% in relation to AERONET, but probably -9-13% to the "truth".
**Reply:**
After the new analysis, which resulted to an overall mean relative bias of -2.7%, this sentence from Section 5 was revised as follows:

*"Additionally, considering the dry bias of the AERONET observations that was discussed in Sect. 2.2, which is about -5 to -10 % (depending on the study and its reference) and varies with season and latitude, it can be concluded that the satellite TCWV observations have a dry bias with respect to the "absolute" truth of about -8 to -13 %, respectively."*

Therefore, the last sentence of the Abstract was also changed to:

*"Overall, the TROPOMI/S5P TCWV product, on a global scale and for moderate albedo and cloudiness, agrees well at -2.7 ± 4.9 % to the AERONET observations, but probably within about -8 to -13% with respect to the "truth"."*

The slope of the linear fit in Figure 6b (Section 4) gives a value of 0.89. So the retrieval actually underestimates by about -10%?
**Reply:** Concerning the slope in Figure 6b, please note that in the process of answering your comments, and after the new analysis that was performed, a new scatter plot was added in Figure 6(b) (see Figure 1 and the respective discussion, below, as a reply to your question about the OLS and TLS methods), for which we applied both the ordinary least squares (OLS) and the total least squares (TLS) methods, to retrieve the respective equations and Pearson coefficients R:

OLS: y=0.9*x+0.9 with R=0.909

TLS: y=1.0*x-0.6 with R=0.904

Considering the OLS equation:

- for a low AERONET TCWV value e.g. 12 km/m$^2$, the respective TROPOMI value would be underestimated by -2.5%.

- for a higher AERONET TCWV value e.g. 45 km/m$^2$, the respective TROPOMI value would be underestimated by -8%.

Therefore, the percentage of the underestimation depends on the magnitude of the "ground-truth". Since, according to the density plot, the majority of the TCWV values lays within 0 – 20 km/m$^2$, the underestimation of -2.7% shown in the histogram (updated Figure 6(a)) is a reasonable result.

L28: Water vapour does not have to form clouds to be transported around the globe.
 **Reply:** The phrase was changed to: "*It is transported through the atmosphere via its circulation and part of the water vapor follows a cycle that consists of cloud formation via condensation, transportation and return to the Earth's surface by precipitation, as rain or snow.*"

L40: It should be mentioned that a major source of stratospheric H$_2$O is methane rather than tropospheric H$_2$O.
 **Reply:** A sentence was added to the manuscript about the effect of the methane and its oxidation on water vapor in the stratosphere. Thank you.
 *"Furthermore, the stratospheric water vapor load is significantly determined by methane and its oxidation within the stratosphere (Le Texier et al., 1988; Oman et al., 2008)."*

L43: Here some exemplary instruments (and corresponding papers) should be mentioned. GPS radio occultation is missing in the list.
 **Reply:** Thank you for this comment. We have added the following paragraph with the relevant information:
 "*We mention here the space-born Medium Resolution Imaging Spectrometer (MERIS) retrievals in the near-infrared (NIR) over land surfaces and coastal areas with the Special Sensor Microwave Imager (SSM/I) TCWV retrievals in the microwave spectra over ocean surfaces (Lindstrot et al., 2014); the TCWV retrieval in the visible blue spectral band for the Global Ozone Monitoring Experience 2 (GOME-2) instruments on board the European Organisation for the Exploitation of Meteorological Satellites (EUMETSAT) MetOp satellites (Chan et al., 2020); the EOS Aura Microwave Limb Sounder (MLS) for water vapor product (EOS, 2017); the MODIS (Moderate-resolution Imaging Spectroradiometer) on board Terra and Aqua total column water vapour (Diedrich, et al., 2015); the Japanese Space Agency Greenhouse Gases Observing SATellite (GOSAT) column-averaged dry-air mole fraction water vapour (Dupuy et al., 2016), etc. Furthermore, long-term ground-based observations also exist such as by the Total Carbon Column Observing Network of ground-based, high-*

*spectral-resolution Fourier Transform Spectroscopy instruments (Wunch, et al., 2011); by the ground-based Global Navigation Satellite System, GNSS (Gendt et al., 2004); by the GCOS Upper Air Network, GUAN, radiosondes (Turner et al., 2003) and by the Aerosol Robotic Network, AERONET, sun photometers (Pérez-Ramírez et al., 2014)."*
All references were added to the Reference list of the manuscript.

L50: It is a bit strange to mention Schneider et al. (2020) and not refer to the other TCWV retrievals from TROPOMI by Borger et al. (2020) and Küchler et al. (2021). In particular considering that Borger et al. (2020) also retrieve TCWV in the visible blue.

> **Reply:** We would like to thank the reviewer. This is an important oversight from our side. A new paragraph was added, above the one mentioning Schneider et al., (2020), with the reference to Borger et al. (2020) and a description of their results:
>
> *"Borger et al. (2020) also retrieved TCWV from the same spectral band of TROPOMI/S5P measurements using the two-step Differential Optical Absorption Spectroscopy (DOAS) approach. The product was intercompared to the Special Sensor Microwave Image/Sounder (SSMIS) onboard NOAA's f16 and f17, the reanalysis model ERA-5 TCWV data and ground-based GPS data from the SuomiNet network. It was found that over ocean and under clear-sky conditions the retrieved TROPOMI/S5P TCWV captures well the global water vapor distribution. Over land, the retrieved TCWV was found to be underestimated by about 10 %, especially during boreal summer, which was attributed to the uncertainty of the external input data, hence some recommendations are given for the use of the product (effective cloud fraction <20 % and AMF>0.1)."*
>
> Küchler et al. (2021) and their respective work was also referenced in another paragraph:
>
> *"Another TCWV product retrieved by the Air-Mass-Corrected DOAS (AMC-DOAS) scheme based on TROPOMI/S5P data in the spectral area 688 to 700 nm, was presented by Küchler et al. (2021). The product was compared to ECMWF ERA-5, SSMIS data and the two scientific S5P/TROPOMI TCWV products that were mentioned above, i.e. the TCWV products described and validated by Borger et al. (2020) and Schneider et al., (2020). These comparisons showed that over sea, AMC-DOAS underestimates TCWV with respect to ERA-5 TCWV, by about 2 kg m$^{-2}$, while its agreement to the TROPOMI/S5P TCWV from Borger et al. (2020) is within 1 kg m$^{-2}$ over both land and ocean. Finally, with respect to the TCWV from Schneider et al., (2020), averaged differences of around 1.2 kg m$^{-2}$ were found."*

L70: TROPOMI was launched in October 2017.

> **Reply:** The date was changed.

L80: DOAS: reference missing (e.g. Platt and Stutz, 2008)

> **Reply:** Thank you for noticing this. The reference was added.

L84: Which improvements have been implemented in the spectral analysis and the AMF calculations?

> **Reply:** Several improvements been implemented to the TROPOMI/S5P TCWV retrieval. Compared to GOME-2, the spectral fitting range is optimized for TROPOMI/S5P observations. In addition, the AMF calculation uses the dynamic a-priori profile rather than the conventional climatology approach. Surface albedo used for AMF calculation is retrieved using the GE_LER approach at the water vapor fitting band rather than the OMI LER product which is retrieved at single wavelength. The details of the improvements/optimizations are addressed in the algorithm paper (Chan et al., 2022).

L92: Not really necessary to mention the data format.

> **Reply:** The phrasing was changed, leaving out the information about the data format.

L114: Replace "utilized" by "used"

> **Reply:** The wording was changed in line 114.

L128: Is the reference source now Martins et al. (2019) or Smirnov et al. (2004) and Alexandrov et al. (2009)?

> **Reply:** Thank you for noticing this ambiguous sentence. The text was re-phrased to:
> *"The total uncertainty of sun photometer retrievals was estimated to be less than 10 % (Smirnov et al., 2004; Alexandrov et al., 2009; Pérez-Ramírez et al., 2014). According to Martins et al. (2019), this percentage was expected to be improved with the implementation of the version 3 of the retrieval algorithm (Giles et al., 2019)."*

L131: "coverage of all continents": looking at Figure 2 only North America and Europe are covered well. Please rephrase.

> **Reply:** The sentence was changed as follows:
> *"The extended network of automatic and quality-controlled observations provides very dense (spatially and temporally) coverage of North & South America, Europe, South-East*

*Asia, as well as Western Africa. This fact, in addition to the homogeneity of the retrieval algorithms, are strong advantages in favor of using the AERONET for this validation work."*

L140: How does the "in-house quality control" look like? Please clarify.

> **Reply:** The sentence was changed to clarify the methodology that was followed in the process of our quality-control:

> *"An in-house quality control based on the visual and statistical analysis of the available datasets per station, ensured that only stations with data that fully cover the time period of our study, and which offer observations within an expected range depending on the station's location, are contributing to the ground-based reference dataset. As a result, the number of stations to be used for the validation of TROPOMI/S5P TCWV was reduced to 369."*

> Please note that in the process of revising the manuscript, the ground-based dataset was also updated and the list of stations that are now used as reference numbers 369 stations. Moreover, the South Pole station (lat: -90°) was decided to be excluded from the reference dataset, since it was offering less than two months of observations to the study, in a latitude that no other source of measurement was available.

Section 3: Here it would be more interesting to show an example from another climate zone rather than showing two similar stations. Replace one of the examples with another one (maybe from the northern mid-latitudes, where most of the AERONET stations are located).

> **Reply:** The station of American Samoa (latitude: -14.25°) was replaced with the station of Acqua Alta Oceanographic Tower (AAOT), located at the Northern Adriatic Sea (latitude: 45.31°).

Figure 3b: Is the value shown for the correlation the Pearson's correlation coefficient R or or the coefficient of determination $R^2$?

> **Reply:** Yes, this is the Pearson correlation coefficient R. It was clarified in the manuscript wherever it is used.

And is the linear fit based on ordinary least squares (OLS) or total least squares (TLS)? Since the uncertainties of TROPOMI and AERONET are of comparable magnitude, a TLS might be more appropriate.

> **Reply:** The linear fit shown in Figures 3b and 4b are ordinary least squares (OLS). The TLS methodology was also applied to the per-station analysis (not shown), as well as to the scatter plot showing the overall correlation between satellite and ground-based observations resulting from all available stations and their co-locations, shown in

Figure 6b (also seen below in this document as Figure 1). The dotted lines show the two different approaches for the statistical analysis: the red line is the ordinary least squares (OLS) method and the resulting equation and Pearson correlation coefficients, R, are shown at the bottom right of the figure; the cyan line represents the total least squares (TLS) method and the respective equation and modified R are shown at the upper left corner of the plot. Both methods result in a very similar Pearson correlation coefficient of slightly above 0.9, which shows the good overall agreement between the two datasets. The slopes of the linear fit are also very close, being 0.9 for the OLS and 1.0 for the TLS. The overall offset between satellite and ground-based observations is 0.9 kg/m$^2$ for the OLS and -0.6 kg/m$^2$ for the TLS.

[Figure]

*Figure 1: The scatter plot showing the correlation between all available TROPOMI/S5P TCWV and AERONET co-located observations.*

L165: 0.788 < 0.79

    **Reply**: We considered that rounding 0.788 to 0.79 would be acceptable. The statistics are now updated.

L166: Shown is the monthly mean bias, right?

    **Reply:** Yes, these are time series of the monthly mean relative biases. It was clarified in the text.

L168: The period of 2.5 years is much too short to speak of a high temporal stability, especially if the time series also has some gaps.

    **Reply:** Thank you for the comment. The sentence was rephrased as follows:

    *"The monthly mean relative bias per station (panels a) depends strongly on the ground-based instrument's operation and maintenance, but for the examples shown here they are within ±0.2 %, showing a good agreement between satellite and ground-based*

*observations and a good temporal stability of both sources of measurement for the available dataset spanning 2.5 years."*

L185: With the high number of measured values, the standard error does not provide any additional, relevant information and should therefore be removed from all comparisons.

> **Reply**: Following the updated co-location methodology, the number of co-locations was reduced to about 70.000, so it was considered best to have the standard error shown in the figures.

L196: A slope of 0.89 basically means underestimation of more than 10%? Has a TLS been used? Negative TCWV values in TROPOMI comparison. Please clarify, where they come from.

> **Reply:** The question about the slope and the TLS method was answered in the "Specific Comments" section.
>
> As for the negative satellite values, this was also answered in the "Major issues" section, above.

L207: "temporal stability": see comment above

> **Reply:** Thank you for the comment. The sentence was rephrased to:
>
> *"The NH curve is continuous with no abrupt changes, showing the temporal stability of both sources of measurement, satellite and ground-based, for the 2.5 years of available data."*

L212: "mainly representing the latitude belt 0° to 60° S": redundant information, as only there is only one station in Antarctica.

> **Reply:** The sentence was re-phrased and the highest Southern Hemisphere latitude was changed in various lines of the manuscript. Actually, after the re-evaluation of the stations that took place in this version of the analysis, the highest latitude Southern Hemisphere station available is at -46°S.

Table 2: What has been the reason for the different latitude binning?

> **Reply:** The reasoning for the latitude binning used in Table 2 is based on the discussion that preceded, according to which, the latitude belts outside the tropics (especially over 45°) have a low water vapor content and variability, therefore, if the binning was done with a 15° step, the differences between the belts 60°N-75°N and 75°-90°N would be negligible.
>
> The statistics in Table 2 were updated and a sentence giving this reasoning was added to the manuscript.

L265: "The performance mainly on the aspect of the surface albedo parameter credibility appears to be sound.": Considering that the albedo is an elementary input parameter of a satellite retrieval, the performance should not be called "sound". Rather acceptable?

> **Reply:** The sentence was changed to:
>
> *"Howbeit, the performance of the TROPOMI/S5P TCWV retrieval algorithm, with respect to the surface albedo parameter which significantly changes with latitude, is currently adequate but could be further improved in the future."*

L285: Instead of looking at the VZA dependence, it might be better looking at the row dependence so that one can see if a West-East dependence exists in the TROPOMI swath.

> **Reply:** A figure with the dependence of the relative differences on the satellite pixel replaced the VZA dependency, as suggested, and the discussion was changed accordingly. No systematic east-west dependence is seen.

L298ff: Please clarify how these dependencies compare to the findings in Borger et al. (2020)?

> **Reply:** The paragraph with the analysis of the dependence on cloud top pressure was enriched with the comparison of our results to those of Borger et al. (2020):
>
> *"For cloud top pressures (panel a) up to 800 hPa, the data bins with relatively high number of co-locations have a positive bias of ~ +5 to +10 %, which decreases to -20 % when the pressure increases to ~900 to 1000 hPa, hence for clouds of lower height that may also affect the ground-based measurements. Borger et al. (2020), that validated their TCWV product against SMISS on board f16 and f17, ERA-5 and GPS data, examined the dependence of their comparisons on cloud height. They also found that low clouds, located below 3-4 km, cause an underestimation in the retrieved TCWV of about -13 %. Typically, the cloud top pressure of 800 hPa that we found to be the turning point, corresponds to ~ 2-3 km, therefore our results are very consistent to Borger et al. (2020), especially considering the fact that they are based on a different retrieval algorithm."*

L315ff: So why do you investigate surface pressures lower than 900hPa, when you only have a limited number of measurements?

> **Reply:** The surface pressure range in only mentioned in the following sentence: "As expected, the bins with pressures less than 900 hPa have a limited number of co-locations and the curve represents mostly noise data.", which is just a comment on what is seen in the figure.

L318ff: Since there are hardly any measurements for albedos > 20%, this should also only limit to 20%, but make the binning finer.

> **Reply:** Thank you for your comment. The figure was changed as suggested and the respective paragraph was re-phrased to:
>
> *"As the density of ground-based stations is much higher at the mid-latitudes of both hemispheres, very few co-locations have surface albedo above 0.2 and since they showed no apparent systematic dependence on surface albedo, they are not included in the figure. For surface albedo values below 0.2, the relative differences range within ±10 %, but no systematic dependence is detected."*

L360: "10-19%": this is only valid for TCWV in the tropics. Likely much higher in the mid-latitudes.

> **Reply:** Indeed, these percentages refer to the tropics and it was mentioned in the first bullet point of Section 5. Unfortunately, we found no publications about the uncertainties in the mid-latitudes.

L375ff: Since there are almost no measurements in the polar regions at high latitudes, the statistics are not meaningful.

> **Reply:** The South Pole station was removed from the list of stations that provide reference data, due to the limited data availability. So the new analysis did not include data for the high Southern latitudes. The sentence was removed.

L382: "cloudiness" → cloud top height

> **Reply:** The wording was changed in line 382.

L386: "temporally stable": see comment above
L386f: "product of high quality and precision, temporally stable and not affected by any other parameters, except from clouds": This is a contradiction in terms, considering that a few lines earlier it is mentioned that the TCWV retrieval likely underestimates by 10-13%, the time series is far too short for analyzing temporal stability and clouds are by far the most important input parameter in satellite retrievals.

> **Reply:** Thank you for your comment.
>
> The issue of the temporal stability refers to the time period of available data only, and that is clarified in various parts of the text, but it will be clarified here as well.

As for our closing statement, in our experience in validating satellite products, as well as according to literature, there is no "perfect" retrieval algorithm. The purpose of a validation paper is to clearly state the advantages and disadvantages of each algorithm and its retrieved product. The fact that the TROPOMI/S5P TCWV product that is under validation has a dry bias is also known from other studies (see Section 2.1). Furthermore, the dry bias seems to be a common feature of TROPOMI/S5P TCWV products retrieved by different algorithms, as it results from other studies cited in our manuscript (e.g. Borger et al. (2020), Schneider et al. (2022) and Küchler et al. (2021)).

Nevertheless, we recognize the need to change the wording in this final paragraph, which is re-phrased as follows:

*"To conclude, as shown from the validation of 2.5 years of available satellite observations, with respect to ground-based observations from AERONET, the TROPOMI/S5P TCWV product retrieved from the blue spectral range, is a temporally stable product of high quality and precision, especially at the tropics. Also, it is not significantly affected by any other parameters, except from clouds when and if some cloudiness at lower atmospheric layers is present in the measurement field."*

L403: Competing interest: To the best of my knowledge, DL is an editor of AMT, which, according to journal's publication guidelines, should also be mentioned here.

**Reply**: Thank you for pointing this out. The following statement was added in the competing interest field:

*"At least one of the (co-)authors is a member of the editorial board of Atmospheric Measurement Techniques"*

If needed, we will consult the AMT editorial office on this matter and follow their instructions.

L448: Apparently, the reference from Kleipool et al. was mixed with another from Köehler et al.

**Reply**: Thank you for noticing this. The reference of Kleipool et al. was corrected.

**Thank you very much for your constructive feedback and your questions.**
**The authors**

---

## Author Comment (AC2)

**General Comments**

This manuscript presents a validation study of the new TROPOMI Total Column Water Vapor (TCWV) retrieved from the 435-455 nm wavelength region. Validation involves comparing 2.5 years of data with AERONET Level 2 precipitable water vapour measurements. Comparisons are performed globally and in several zonal bands to determine the mean bias. The impact of viewing geometry (solar zenith angle, viewing zenith angle), cloud properties (cloud top pressure, cloud albedo, cloud fraction), and retrieval parameters (surface pressure, surface albedo, air mass factor, fit RMS, water vapour) on the comparisons are also examined.

The analysis is straightforward and provides a useful contribution to the evaluation of TROPOMI water vapor. However, the paper would be strengthened by making greater use of the validation dataset, examining the differences across the 351 stations rather averaging across zonal bands. Overall, I recommend publication after the comments below are addressed.

Thank you for recommending our paper for publication.
* * *
**Specific Comments**

**Page 2, second last paragraph:** The previous paragraph mentions the TROPOMI SWIR TCWV data product and its validation against TCCON. Why wasn't this new TROPOMI TCWV product compared against the SWIR product? Why wasn't TCCON included, or other available water vapour datasets such as GPS/RO? Explain why AERONET was chosen as the comparison dataset for this validation study.

**Reply:** We thank the reviewer for this comment.

The main reason for basing our work on AERONET water content observations is that, first of all, the network is very well established, with more than 25 years of operations and a transparent data quality assurance plan through its extensive cal/val routine operations, see here: System Description - Aerosol Robotic Network (AERONET) Homepage (nasa.gov). Furthermore, the network offers a complete global coverage, covering the entire planet quite satisfactorily, see here: AERONET Data Display Interface - WWW DEMONSTRAT (nasa.gov) .

Even though it is true that AERONET has been extensively used for AOD validation, the AERONET water vapor observations have also been employed in space-born and ground-based instrumentation validation studies, see for e.g.

https://doi.org/10.1016/j.atmosres.2019.04.005, https://doi.org/10.3390/rs13163246, https://doi.org/10.5194/amt-11-81-2018, etc. Due to the fact that a number of studies have already utilized the ground-based GPS datasets, and the same TCWV from TROPOMI/S5P product was also validated against GNSS (https://doi.org/10.3390/atmos13071079), in this work we aimed to start with the AERONET TCWV. Your suggestion is of course very welcome for future works.

**Page 6, line 147:** Define the equation used to calculate percentage difference, e.g,. 100 x (TROPOMI – AERONET) / AERONET, so that this is clear. Also, this quantity should be called the relative difference. Is "minimize the noise" the best description? The choice of coincidence criteria is a trade-off between maximizing N for better statistics and minimizing space and time differences between the comparison datasets. No real justification is given for the choice of 10 km and 30 minutes; have these values been used in other water vapour validation studies or were trade-off curves constructed to find the optimum criteria? Do the comparisons involve single or multiple pairs, i.e., is each TROPOMI measurement compared with the closest AERONET measurement (or vice versa), or are multiple comparisons allowed the space and time criteria are met for a TROPOMI measurement and multiple AERONET measurements (or vice versa)?

> **Reply:** The equation used to calculate the percentage difference was added in Section 3 and the manuscript was changed to refer to it as relative (percentage) difference.
>
> The choice of the coincidence criteria was indeed the outcome of some studies we made in our effort to find the best trade-off between the number of co-locations and the results of their statistical analysis, taking into account the characteristics of the instruments, satellite and ground-based.
>
> - It should be noted that the 10 km maximum search radius was only used for the extraction of the overpass files, taking under consideration the high spatial resolution of TROPOMI/S5P observations ($3.5 \times 7$ km$^2$ until Aug. 2019 and $3.5 \times 5.5$ km$^2$ thereafter). Other studies, such as Borger et al. (2020), Xie et al. (2021) etc. used a similar distance for their validation work with respect to ground-based measurements. Besides, our aim was to use only the closest co-locations in space and in time for our statistical analysis, which leaded to a maximum spatial difference between the ground-based station and the satellite instrument of up to 5 km, as it is illustrated in Figure 1 of this document.
> - As for the very strict criterion of up to 30′ temporal difference between the satellite and ground-based observation, which is a much smaller time window than what other studies have used (for example, Chan et al. (2020) and Borger et

al. (2020) allow up to 2 hours, while Xie et al. (2021) also uses a 30′ temporal difference), it was based on the fact that the AERONET dataset provides clear-sky measurements only, resulting to rather invariable temporally observation field.

[Figure]

*Figure 1: The dependence of the relative difference between satellite and ground-based observations on the distance of their co-location. Each one of the gray data points consist of less than 3% for the total number of co-locations.*

Please, note that in the process of the revision of our manuscript, it was found that a full re-analysis of our validation work had to be done. When the first validation exercises were performed, based on a limited dataset that was available at the time, we used the following co-location technique to increase the sample of the data to be evaluated:

Each satellite measurement from a specific pixel (the one spatially closest to the ground-based station within 10km), was compared to all instantaneous ground-based measurements that were performed within a ±30′ temporal interval with respect to the time of satellite observation. This way, for each overpass instance there were more than one matching pairs, explaining the high number of co-locations.

This approach is not necessary now that a significantly larger satellite dataset of 2.5 years, is available. Therefore, the co-location methodology was changed to keeping only the match with the minimum temporal difference between satellite and ground-based observations within a 10 km radius, if this temporal difference is up to 30 min. The resulting total number of co-locations is now about 70.000. The new methodology was applied throughout the manuscript and all plots and statistics were updated and the manuscript was revised accordingly. Additionally, a paragraph with a detailed description of the co-location methodology, giving all the above-mentioned details, is added in Section 3.

**Page 7, line 156:** The 633,000 coincident measurements constitute a rich validation dataset that could be investigated in more detail. This large number would seem to be the justification for using AERONET for the validation, but the analysis doesn't take full advantage of the resulting information. Since a per-station analysis has already been done, there are 351 global comparisons – it would be interesting to examine these and to look more carefully for spatial differences and dependencies. For example, consider adding a panel to Figure 8 that shows the seasonal and latitudinal variability of the mean bias, and a similar figure showing latitudinal and longitudinal variability, using the results from all 351 stations. Another panel that could be added to Figure 8 is the seasonal and latitudinal variability of N, given the discussion on lines 228-233.

> **Reply:** Thank you for the comment and your suggestions.
>
> - A panel with a new contour plot was added in Figure 8, showing the seasonal and latitudinal variability of the mean bias, as it was suggested.
> - Following the suggestions of Reviewer #1, the latitudinal and longitudinal variability of the mean relative bias was presented in the form of a world map, showing the relative mean bias of each station using a colorbar. To further investigate any possible patterns in relative bias over Europe and N. America, where the stations are very dense, the two areas were also plotted separately. The three maps (world, Europe and N. America) were added in Appendix A and commented in Section 4.
> - As for the seasonal and latitudinal variability of the number of co-locations, we believe that the information was already given in the timeline plot shown in Figure 5.

**Page 7, line 160 and Figures 3 and 4:** State why monthly mean percentage differences are calculated and plotted (perhaps to provide more even annual coverage?). If the mean bias and standard deviation are calculated using all individual points for a station, do the results differ from those obtained using the monthly means?

> **Reply:** The use of monthly means is adopted only when time series are shown, to keep the figures clear and to be able to detect any possible seasonal variability. For example, when a pole-to-pole graph is made, all individual (instantaneous) co-locations within each latitude belt are considered and, of course, they are not temporally averaged.
>
> As for the difference in the statistics when using monthly means with respect to individual co-located data, you see in the example below (Figure 2 of this document), where:

- to the left, all individual co-locations for the station of Santa Cruz, Tenerife, are plotted in the form of time series, and
- to the right, the time series of the monthly means are depicted,

that the main difference between the two ways of illustration, as it is expected, is the standard deviation of the mean, which is almost double when all individual points are used for the statistics.

[Figure]

*Figure 2: The time series of the relative differences between the co-located satellite and ground-based TCWV observations, in the form of raw, instantaneous percentage differences(left panel) and monthly means of the instantaneous differences (right panel), for the station of Santa Cruz, Tenerife.*

**Page 7, line 167:** "The mean relative bias per station (panels a) depends strongly on the ground-based instrument's calibration," Since all AERONET data is Level 2.0, what calibration issues are there? Nothing has been said about this elsewhere in the paper so this should be explained. Is there a parameter defining the calibration status for each station so that the dependence of the mean relative bias on the calibration can be determined?

**Reply:** We thank the reviewer for this comment. The wording used in the text does not reflect what we really meant. Of course, using Level 2.0 excludes any calibration issues from the discussion. The sentence was rephrased as follows:

*"The monthly mean relative bias per station (panels a) depends strongly on the ground-based instrument's operation and maintenance, …"*

**Technical Corrections**

Page 1, line 9: here and elsewhere throughout the paper, delete "very". It is used too frequently and is not needed.

**Reply:** The use of the word "very" is minimized throughout the manuscript.

Page 1, line 11: (435-455 nm)

**Reply:** Corrected.

Page 1, line 15: although AERONET has 1300 stations, data from only 351 are used in the study – this should be noted here

**Reply**: The number of stations was added in the sentence.

Page 1, line 18: "of the order of only -3% for the mid-latitudes and the tropics (+-60deg)" does not seem consistent with the mean bias numbers in Table 2 which are -4.0, -5.9, -9.6, -5.9, 2.4, and 5.8 for the six bands between 60N and 60S, nor with the NH, SH, and global biases in Table 1. Provide more specific results here.

**Reply:** This percentage referred to the mean relative bias for the mid-latitudes and the tropics and resulted from averaging the mean relative biases of all latitude belts within ±60°.

Table 2: The zonal statistics of the co-located satellite and ground-based observations

| Hemisphere | Latitude belt | Mean Diff.[1] (kg/m$^2$) | Mean Rel. Bias (%) | Mean St. Dev. (%) | Mean St. Err.[2] (%) |
|---|---|---|---|---|---|
| NH | 90°-60° | -0.4 | $1.2 \pm 31.5$ | 61.3 | 12.6 |
| | 60°-30° | -0.8 | $-4.0 \pm 2.9$ | 44.0 | 1.3 |
| | 30°-15° | -2.2 | $-5.9 \pm 3.4$ | 23.6 | 1.6 |
| | 15°-0° | -3.7 | $-9.6 \pm 3.0$ | 18.5 | 2.0 |
| SH | 0°-15° | -2.5 | $-5.9 \pm 5.5$ | 32.2 | 3.3 |
| | 15°-30° | -0.7 | $2.4 \pm 8.3$ | 52.3 | 3.6 |
| | 30°-60° | +0.5 | $5.8 \pm 12.3$ | 46.1 | 8.9 |
| | 60°-90° | +0.3 | $42.2 \pm 4.9$ | 84.8 | 16.5 |

[1] Satellite-Ground

[2] 99.7% CI

The abstract was revised and now the sentence refers to the overall mean relative bias, which is -2.7 %, after the new analysis that followed the revision of the co-location methodology:

*"The Pearson correlation coefficient of the two products is found to be 0.91 and the mean bias of the overall relative percentage differences is of the order of only -2.7 %."*

Please note that the statistics in Table 2 were updated.

Page 1, line 19: delete "influence"
   **Reply:** Deleted.

Page 1, line 21: define CTP, clarify what "low cloudiness" means – low cloud top pressure?
   **Reply**: The sentence was rephrased.

Page 1, lines 25-29 and elsewhere in the manuscript (lines 33, 70, 72, 208, etc.): change "earth" to "Earth" (the planet) throughout
   **Reply**: Corrected.

Page 2, line 31: delete "very"
   **Reply**: Deleted.

Page 2, line 34: remove/replace one of the "therefore"s
   **Reply**: Corrected.

Page 2, line 37: delete line break
   **Reply**: Deleted.

Page 2, line 38: delete "very"
   **Reply:** Deleted.

Page 2, line 40: high-latitude
   **Reply**: Corrected.

Page 2, line 41: delete "key" (already say "important")
   **Reply**: Deleted.

Page 2, line 41: "for the evolution of the greenhouse effect and the projection …"
   **Reply:** Corrected.

Page 2, line 41: climate change
   **Reply:** Corrected.

Page 2, line 45: (435-455 nm)
   **Reply**: Corrected.

Page 2, line 45: delete "further"
>**Reply**: Deleted.

Page 2, line 47: delete "sensors"
>**Reply**: Deleted.

Page 2, lines 50, 54: clear-sky
>**Reply**: Corrected.

Page 3, line 66: delete "influence"
>**Reply**: Deleted.

Page 3, line 70: TROPOMI was launched on 13 October 2017
>**Reply**: Corrected.

Page 3, line 79: (435-455 nm)
>**Reply**: Corrected.

Page 3, line 81: "in short, a two-step approach …"
>**Reply**: Corrected.

Page 3, line 84: air mass factor
>**Reply:** Corrected.

Page 3, lines 95-96: the seasons listed here (winter, spring, summer, autumn) only apply to the Northern hemisphere – either add NH before each season or remove the seasons from this sentence.
>**Reply:** The seasons were removed from the sentence.

Page 4, line 99: "decreasing below 5-10 kg/m2 closer to the poles."
>**Reply:** Corrected.

Page 5, line 116: product
>**Reply:** Corrected.

Page 5, lines 130-131: for the 351 stations used in this study, AERONET coverage of all continents is not actually "very dense" spatially as seen in Figure 2 – revise this description.

**Reply:** The sentence was changed as follows:

*"The extended network of automatic and quality-controlled observations provides very dense (spatially and temporally) coverage of North & South America, Europe, South-East Asia, as well as Western Africa. This fact, in addition to the homogeneity of the retrieval algorithms, are strong advantages in favor of using the AERONET for this validation work."*

Page 5, line 131: delete "very"

**Reply:** Deleted.

Page 6, line 140: "resulted in the reduction ..." Describe the in-house quality control that reduced the number of stations with usable data from 1300 to 351.

**Reply**: The sentence was changed to clarify the methodology that was followed in the process of our quality-control:

*"An in-house quality control based on the visual and statistical analysis of the available datasets per station, ensured that only stations with data that fully cover the time period of our study, and which offer observations within an expected range depending on the station's location, are contributing to the ground-based reference dataset. As a result, the number of stations to be used for the validation of TROPOMI/S5P TCWV was reduced to 369."*

Please note that in the process of revising the manuscript, the ground-based dataset was also updated and the list of stations that are now used as reference numbers 369 stations. Moreover, the South Pole station (lat: -90°) was decided to be excluded from the reference dataset, since it was offering less than two months of observations to the study.

Page 6, line 142: "as can be ..."
**Reply:** The sentence is deleted.

Page 6, Figure 3 and page 7, Figure 4: The quality of the fonts is poor on these panels and hard to read – the fonts should be improved.
**Reply:** The figures were replaced by new ones of better quality.

Page 7, line 160: Why show monthly mean percentage differences
**Reply:** This was answered above, in the section of the Specific Comments.

Page 7, line 163: delete "very nice" (here and elsewhere, these subjective descriptions can be removed)

**Reply:** Deleted.

Page 7, line 164: state whether the correlation coefficient is R or R^2

**Reply:** This is the Pearson correlation coefficient. It was clarified in the manuscript.

Page 7, lines 165 and 167: delete "very"

Reply: Deleted.

Page 8, Figure 5: The quality of this figure is poor; at a minimum, the y-axis should be extended beyond +-90 so that the highest latitude points are visible, the fonts should be improved, and "AERONET" removed from the top.

**Reply:** The figure was changed according to the reviewer's suggestions. The y-axis range was not changed beyond ±90° because in the process of the re-evaluation of our co-located data, the South Pole station was decided to be left out due to its limited temporal coverage (2 months of observations in 2018).

Page 8, line 180: 60S is mentioned here (and again on line 212) but line 177 says that the SH data only extend to 55S – should 60 be changed to 55?

**Reply:** Yes, the latitude was changed in both lines. Actually, after the re-evaluation of the stations that took place in this version of the analysis, the highest latitude Southern Hemisphere station available is at -46°S.

Page 8, line 183: averaged

**Reply:** Corrected.

Page 9, line 191: on a global scale

**Reply:** Corrected.

Page 9, line 192: change "about" to "approximately"

**Reply:** Corrected.

Page 9, line 195: correlation coefficient R?  delete "very"

**Reply:** Yes, this is the Pearson correlation coefficient. The information was added to the text.  "Very" was deleted.

Page 10, Figure 7: the legend (TROPOMI) on the lower left of the panels and "AERONET" in the lower right should be deleted.  State in the figure caption what the errors bars are.

**Reply:** The figures were changed according to the reviewer's suggestions. A sentence is added in the figure caption to explain the error bars.

Page 11, Figure 8: delete "TROPOMI" and "AERONET" from the panel

**Reply:** The figure was changed according to the reviewer's suggestions.

Page 12, lines 235, 236, 245, 252: delete "very", etc.

**Reply**: Deleted.

Page 12, line 253: 80deg S to 90deg S

**Reply:** Corrected.

Page 12, line 261: location

**Reply:** Corrected.

Page 12, line 264: regarded as [very] good.

**Reply**: Corrected.

Page 12, line 265: rewrite this sentence for clarity – it is not clear what is meant

**Reply:** The sentence was changed to:

*"Howbeit, the performance of the TROPOMI/S5P TCWV retrieval algorithm, mainly on the aspect of the surface albedo parameter, which significantly changes with latitude, is adequate and could be further improved in the future."*

Page 13, line 273: delete "influence", change "quantities" to parameters or variables

**Reply**: The term "influence quantity" was replaced by "parameter"

Page 13, line 274-275: "detailed results" is a strange term – how is air mass factor a detailed result?  Are these outputs from the retrieval algorithm?

**Reply**: The sentence was changed as follows:

*"These quantities can be parameters that are used as inputs for the TCWV retrieval algorithm, such as cloud and surface information, or algorithm-related parameters, like the air mass factor."*

Page 13, line 276 and Figures 10-13: the numbers at the top of each panel are unreadable – revise these plots to show this information in another way.

**Reply:** The figures were changed and we used the following sentences in Section 4.2, 1$^{st}$ paragraph, to clarify the new way of illustration:

"*Note that, in the following figures, when the number of co-locations that are averaged for each bin is less than 3% of the total, the respective the data point is shown in gray (instead of blue). This is a way to distinguish the data points in terms of relative importance.*"

Page 13, line 276: change "of each figure" to "Figures 10-13"

**Reply:** Due to re-phrasing of the sentence (see above) this correction is not needed anymore.

Page 13, line 280: specify whether the SZA and VZA are for TROPOMI or AERONET. Define what is meant by the viewing zenith angle.

**Reply:** These are parameters coming from the TROPOMI dataset and it was clarified in the manuscript. Following the suggestion of Reviewer #1, the VZA dependence figure was replaced with another one showing the dependence on the satellite pixel, to investigate if a West-East dependence exists in the TROPOMI swath. Therefore, a definition is not needed now.

Page 13, line 281: delete line break

**Reply:** This is probably an issue of different word versions used, as in our document no line break is included at that location. If you mean that the first sentence of the section and the following paragraph should be joined, that was done.

Page 13, line 285: it's not clear what is meant by "the dependence of the percentage differences on SZA is ~13%". The differences in Figure 10(a) vary from approximately -10% to +10% so where is 13% coming from? Has a line been fitted to the data? If so, should it be added to the plot? Discussion of these results should be clarified.

**Reply:** Thank you for your comment. The difference (now 15%) that is mentioned here is the peak-to-peak difference seen for the total range of SZAs, since below 45° the co-locations' mean relative biases are negative, up to -6 %, and above 70° they have a mean relative bias of ~ +8%. The sentence was re-phrased as follows:

"*Overall, the dependence of the relative differences on SZA is ~15 % peak-to-peak.*"

Page 13, line 285: "of the mean increases for larger/smaller SZA" "higher SZA" is ambiguous – specify whether larger or smaller

>**Reply:** The word "larger" was used instead of "higher".

Figures 10-14: Improve the presentation of numbers as noted above. Delete the legend (TROPOMI) on the lower left of the panels and "AERONET" in the lower right. State in the figure caption what the errors bars are. Revise the figure captions so that they are consistent between figures and fully describe what is shown in the panels.

>**Reply:** The suggested improvements for the figures were adopted. Thank you.
>An extensive description of what is seen in the plots that follow and how it is calculated (monthly means and error bars) is given in the last paragraph of Section 3. Nevertheless, the information for the error bars representation is also added in the figure captions. All figure captions were carefully reviewed and corrected where necessary.

Page 15, lines 297 and 304-307: discuss the dependence on cloud fraction above 0.3

>**Reply:** The paragraph discussion cloud fraction was changed as follows:
>*"The cloud fraction figure (panel c) shows that the vast majority of the co-locations have cloud fraction values below 0.3, which is expected since both satellite and ground-based observations are filtered for cloudiness (satellite data are filtered for cloud fraction < 0.5). Within the cloud fraction range of 0 to 0.3, no particular dependence is seen. The co-locations that are characterized with cloud fraction between 0.3 and 0.5 are very few in population but they introduce high positive mean relative biases. Filtering-out the co-location dataset for cloud fraction > 0.3 was also investigated, resulting to no major differences in the validation results. Nevertheless, it is advisable to not use this small portion of TCWV data for future scientific studies."*

Page 16, lines 326-327: change the title of this section – "Detailed results" is not informative. Are these outputs from the TROPOMI retrieval algorithm?

>**Reply:** The title of the section was changed to "Dependency on algorithm-related parameters".

Page 16, line 332: from Figure 13(a), the largest negative value for AMFs of 2-4 looks like approx.. 25%, not 18%

>**Reply:** Thank you for the comment. The percentage was changed to 20% according to the updated figure 13(a).

Page 16, line 336: "with a low ..."

>**Reply:** Corrected

Page 16, line 338: delete "result"
> **Reply:** Deleted

Page 16, line 342: areas
> **Reply:** Corrected

Page 16, line 343: this sentence is unclear – what does "they" refer to and what "should be treated with caution? Revise for clarity.
> **Reply:** The sentence was changed as follows:
> *"Even though very few retrievals are based on an AMF value of less than 1, we note that comparisons to these pixels result in extremely high relative differences with respect to ground-based measurements, up to -100 %."*

Page 17, line 346: "namely: TROPOMI (a)…"
> **Reply:** Corrected

Page 17, line 348: is "statistics" needed in the title of this section?
> **Reply:** Section 5 was renamed to "Summary and conclusions".

Page 17, line 350: consistency with
> **Reply:** Corrected

Page 17, line 353: corresponding to clear-sky
> **Reply:** Corrected

Page 17, line 355: summarized as follows
> **Reply:** Corrected

Page 18, line 358: delete "excellent", change "their" to "the"
> **Reply:** Corrected

Page 18, line 359: mean bias is -4.7%
> **Reply:** After the new analysis, the overall mean relative bias resulted to -2.7 %. The hemispherical mean relative biases are now -3.1 % for the NH and +0.9 % for the SH.

Page 18, line 363: should "accuracy" be "consistency"?

**Reply**: Thank you for the comment. The sentence was re-phrased to:

*"The mean standard error of the comparisons, at a 99.7% CI, is 0.5 %, highlighting the consistency of the results. "*

Page 18, line 381: does "low cloudiness" mean low cloud top pressure?

**Reply:** The term "low cloudiness" was changed to "low cloud top height".

Page 18, line 386: Is 2.5 years of comparisons sufficient to claim temporal stability? What is the basis for claiming high precision? Accuracy appears to be -9% to -13% based on line 365. Revise this sentence.

**Reply:** Thank you for your comment.

The issue of the temporal stability refers to the time period of available data only, and that is clarified in various parts of the text, but it will be clarified here as well.

The wording in this final paragraph was is re-phrased as follows:

*"To conclude, as shown from the validation of 2.5 years of available satellite observations, with respect to ground-based observations from AERONET, the TROPOMI/S5P TCWV product retrieved from the blue spectral range, is a temporally stable product of high quality and precision, especially at the tropics. Also, it is not significantly affected by any other parameters, except from clouds when and if some cloudiness at lower atmospheric layers is present in the measurement field."*

The high precision of the product refers mainly to the tropics, where (as seen in Figure 8b) the standard deviation of the mean bias has a very low variability.

Page 18, line 388: delete "very"

**Reply:** Deleted

Page 18, line 389: list other "blue-band satellites"

**Reply:** The following satellites were listed in the sentence:

- Global Ozone Monitoring Experience (GOME) mission (Burrows et al., 1999),
- SCanning Imaging Absorption SpectroMeter for Atmospheric CHartographY (SCIAMACHY; Bovensmann et al., 1999),
- Global Ozone Monitoring Experience 2 (GOME-2; Callies et al., 2000)
- Ozone Monitoring Instrument (OMI; Levelt et al., 2006),

**Thank you very much for your constructive feedbacks and questions.**

**The authors**

---

## Author Response (AR2)

Dear Ass. Editor,

Thank you for your time and effort to go through our revised manuscript. Herein we will try to answer to your comments.

*"One important reason is that you have changed the data significantly with respect to the previous (reviewed) version. If data are changed significantly, the paper needs another round of review."*

> Thank you for pointing this issue, but in fact we did not change the dataset that was used for this validation work. We still use the same TROPOMI/S5P TCWV data and the same Level-2 data from AERONET stations as in the initial version of the paper, but we modified the comparison methodology, following well-justified and valuable suggestions from the reviewers. Namely, instead of comparing all CIMEL observations within ±30' from the satellite overpass time, in the revised version we considered only the nearest in time (within a ±30' time interval) ground-based observations. This change led to similar results concerning the relative bias but reduced the dispersion of the comparisons.

*"Second, although your statistical analysis appears sound, no attempt is made to explain the results phenomenologically. For example, the patterns in Fig. 8 are not explained using meteorological or observational variability. In fact, in the description of Fig. 9 the latitudinal dependence of the relative difference is deemed non-existent ("no clear pattern in the dependency of the relative differences on latitude").*

> Validation studies can be a multi-phase activity, involving the interaction between the data producers and the validation team. It is common practice, as a first step, to use the comparisons to a dataset characterized as reference to investigate and identify issues related to the algorithm and observing geometry of the satellite product. This step uses global databases. Then at a second step, when certain instrumental issues are excluded, attempts are made to understand the origin of remaining differences, in collaboration with the algorithm teams. Indeed, to that respect, case studies and association of the observed differences with the variability of geophysical variables that affect the retrievals are extremely useful, but these will require more analysis which could be part of a follow up paper.

> To explain better the patterns seen in Fig. 8, we changed the manuscript. Please see our reply to your comment #14, below. For the discussion on Fig. 9, please see our answers to your comments #17 and #18, below.

*"Moreover, whereas the other TROPOMI TCWV retrieval validation papers are now referenced and the results compared with those from your own validation, similarities and differences between the algorithms are not discussed. These would be of use to understand why validation results agree - or not."*

> Concerning the differences between Chan et al. (2022) and the other two retrieval algorithms for TROPOMI TCWV products (Borger et al., 2020; Schneider et al., 2020

and 2022), they are now added to the manuscript (Section 1), as suggested. See also our answer to comment #2, below.

*"Third, there are a lot of qualitative statements ("the performance of the TROPOMI/S5P TCWV retrieval algorithm (...) is currently adequate but could be improved further in the future."; "is a temporally stable product of high quality and precision..."). The findings of a validation study should be given in quantitative terms, e.g., "The product shows a stability of X%/decade, the uncertainty range is Y, and the mean bias Z; this is (or is not) in agreement with requirements / in agreement with previous studies.".*"

In such an extensive article, the use of qualitative terminology is indeed unavoidable at points during the discussion. We share the editor's view that numerical findings should be clearly stated in a paper, which is why we opted to include numerical results both in the discussion of the figures and statistical results as well as a bullet-point presentation in the conclusions.

According to the co-authors' experience in the field of satellite products' validation and the respective literature, apart from quantitative information, general statements are useful for a potential user of the satellite data to put these quantitative statements in perspective. Therefore, we indeed include a few such statements like the two that you point out. Specifically, the second sentence (*"is a temporally stable product of high quality and precision..."*) is part of the paper's concluding remarks and follows one full page of conclusions (Section 5) which is rich in numbers and percentages.

*"And fourth, the arguments in this manuscript are often not properly backed by evidence. Example: "The monthly mean relative bias (...) depends strongly on the ground-based instrument's operation and maintenance". I doubt it - but if it is true, there should be a reference to literature or other reliable source."*

Specifically for this sentence we give extended answers in your comments #11 and #15, below.

After taking under consideration your valued questions, comments and suggestions, our manuscript was revised again, and a new version is submitted.

We would like to sincerely thank you for your help in the process of improving our manuscript and your constructive comments.

The authors

**Replies to specific comments on the manuscript:**

(Please note that the numbering of the lines seen below corresponds to the manuscript version that you used to provide your comments/questions and not the updated version of the manuscript that we submit after your comments.)

1. Abstract, line 28: "This is quite a significant change wrt the old version"

   **Reply:** Indeed, the mean relative bias was decreased by 1.3% with respect to the previous version, still within the 1-sigma statistical variability. This decrease is due to the change in the co-location methodology that was applied after the implementation of the reviewers' suggestions.

2. Introduction, lines 68-81: "Please describe the differences between Chan, Borger, and the other TROPOMI algorithm so that we can relate those to the agreements in validation results"

   **Reply**: The following sentences will be added at the end of this paragraph (line 75):

   *"The methods of Borger et al. (2020) and Chan et al. (2022), are similar in principal but they differ in some important aspects such as: (i) Chan et al. (2022) fit for the 435-455nm spectral range, while Borger et al. (2020) use a slightly different wavelength range, 430-450nm; (ii) for the AMF calculation, the algorithm of Borger et al. (2020) assumes an exponential decay profile with empirical parameterization of the water vapor scale height, while Chan et al. (2020) use an a-priori profile from the statistical analysis of historical data and they dynamically pick the most appropriate one; (iii) for the surface albedo parameter, Chan et al. (2020) use the TROPOMI/S5P GE_LER which is derived at the same spectral fitting range (435-455nm), while Borger et al. (2020) use the OMI surface albedo retrieved at 442nm. The comparison of the two surface albedo products is extensively discussed in Chan et al. (2022)."*

   The Schneider et al. algorithm is using a different spectral range, which is already mentioned in the manuscript (lines 76-77). The following sentence will be added after the respective paragraph (line 81):

   *"Compared to Chan et al. (2022), the Schneider et al. (2020, 2022) algorithm employed a completely different technique and due to the differences in the spectral range of the measurement, the final water vapor product has a different vertical sensitivity."*

3. Introduction, line 80: "Why did you make a change here? This is not a new result, right?"

   **Reply**: Actually, no, this is not a new result. The percentages given in the previous version of the manuscript referred to mid-latitude stations only. I deleted the phrase "for mid-latitude stations" and changed the percentages according to Schneider et al. (2020) to give a more general overview of the product's bias.

4. Section 2.1, line 126: "Please keep units consistent!"

   **Reply**: These are the units that Vaquero-Martinez et al. (2022) use in their work. But we will, of course, turn them into $kg/m^2$.

5. Section 2.2, line 154: "And should therefore not affect your results, since you are comparing clear-sky data only, right?"

**Reply**: Cimel will measure TCWV when the solar disk is clear, but this does not mean that there are no clouds within a radius of 10 km around the station. This is why the cloud fraction of the satellite measurements (Figure 11) goes up to 0.5. Therefore, we are not comparing only clear-sky data.

6. Section 2.2, line 165: "So which is it??? Can you combine all these numbers into a single number?"
   **Reply**: The uncertainty of the AERONET v3 algorithm is not assessed by any of these papers, but as it is written in line 166, it is expected to be better than 10%, which is the uncertainty of the v2 algorithm (line 165). As for the AERONET dry bias, it is summarized in the phrase: "-5 to -10% depending on the study and its reference" (line 458, Section 5). The following sentence will be added in this paragraph (line 162, before Campanelli et al.), as well:
   *"An approximate mean bias for the AERONET TCWV product that results from all these studies, that are based on various stations, temporal coverages and reference measurements, is -5 to -10 %."*

7. Section 2.2, line 171: "...use of a single retrieval algorithm..." That, and the consistency in instruments and calibration!
   **Reply**: Thank you for the suggestion, this sentence will be rephrased to:
   *"This fact, in addition to the use of a single standardized retrieval algorithm and the consistency in instruments calibration, are strong advantages in favor of using the AERONET for this validation work."*

8. Section 2.2, lines 180-183: "This is neither transparent, nor reproducible. You need to put quantitative information here. How many months of data were required, etc."
   **Reply**: The phrase will be changed to:
   *"An in-house quality control based on the visual and statistical analysis of the available datasets per station, ensured that only stations with data that cover fully the time period of our study, or cover at least 20 out of the 32 months of the TROPOMI/S5P dataset, are contributing to the ground-based reference dataset."*

9. Section 3, line 191: "Not a very scientific argument. Better would be: considering the scale of spatial variability of TCWV, the value detected at 10 km distance is expected to be very close."
   **Reply**: Thank you for the comment. In this sentence, we are only illustrating that we used the same criteria for the co-locations as other studies did. We think that what you suggest is already stated in the previous sentence.

10. Section 3, line 198: "I think you mean continuous. You are using Level-2 data, which is not at all instantaneous."
    **Reply**: Actually, no, the term "instantaneous" does not mean it is near-real-time (which, of course, Level-2 data are not). As we explain further in the sentence, we mean that these are not hourly or daily mean values. The record used here consists of all individual measurements performed every 15' during each day. To further clarify this issue, we will also add this information in line 177, where the term in traduced for the first time:

*"The data files retrieved from AERONET are available in ASCII format in daily, monthly or instantaneous (i.e. measurements performed every 15') temporal analysis"*

11. Section 3, line 216: "You argued before that the instruments are very similar (as a reason to use AERONET), so this cannot be right. Also, I wonder how you come to this conclusion."

    **Reply**: Thank you for the comment. You are right, this phrase does not communicate correctly the message that we had in mind. The sentence will be rephrased to:
*"The monthly mean relative biases per station (panels a) for the example stations shown here are within ±0.2 %, demonstrating the good agreement between satellite and ground-based observations, as well as a good temporal stability of both sources of measurement for the available dataset spanning 2.5 years. The variability of the biases, depicted as error bars, may be due to both the ground-based and space-born instrument observational accuracy as well as algorithm and/or meteorology-related effects."*

12. Figure 3: "The figure has changed quite a bit"

    **Reply**: Yes, it has changed due to the different co-location methodology that was applied following the suggestions of the reviewers.

13. Section 3, lines 235 – 237: "And no TROPOMI data..."

    **Reply**: TROPOMI/S5P data are available in the South Pole. We did have co-locations for a two-month period during local summer of 2018-2019 (mid-November until mid-January), but it was decided to not use them in the analysis, due to their lack of representativeness.

14. Figure 8: "Are you going to explain these patterns?"

    **Reply**: The discussion on Fig. 8 was further developed and the patterns were linked to the latitudinal and temporal changes in TCWV content seen in Fig. 1 and Fig. 9. The final form of the paragraph is quoted after your comment # 18, below.

15. Section 4.1, line 346: "What does this mean? How do you know? Literature reference?"

    **Reply**: This was a wrong choice of words and thank you for noticing. The sentence will be rephrased as follows:
*"…since it is well known that some ground-based stations may overestimate or underestimate their observational constituent systematically due to the meteorological conditions occurring at the station site. Moreover, when such a station does not provide a continuous record of observations, there is a high possibility that it will introduce an artificial and non-representative bias to the validation. Most of these stations, that did not fully cover the time period of our study, were filtered out of the ground-truth database used in this work."*

16. Section 4.1, line 350: "To me, co-location statistics are the number of overpasses, or similar. I don't think that's what's meant here."

    **Reply**: Thank you for the comment. The phrase will be changed as follows:
*"... the respective stations were considered with the remark that the statistics resulting from their co-locations should be interpreted with caution."*

17. Section 4.1, line 351: "Yes there is - see Fig. 8"
    **Reply**: Of course, you are right! This phrase ("Nevertheless, as shown above, ...... overestimation close to the poles.") will be deleted. Moreover, since the dependency of the percentage differences on latitude is already discussed in the previous paragraph, discussing Fig. 9, the next two sentences ("Considering the uncertainties … in the future.") will be moved to the end of the previous paragraph. The final form of the paragraph is quoted after comment #18, below.

18. Section 4.1, lines 354-356: "I do not know what this conclusion is based on."
    **Reply**: This conclusion is based on the previous sentence, that compares the measurement uncertainties to the mean biases per latitude bin showed in Fig. 9. To make it clearer and to also introduce the dependence of the product on surface albedo, the sentence will be changed as follows:
    *"The performance of the TROPOMI/S5P TCWV retrieval algorithm with respect to the surface albedo parameter which significantly changes with latitude is currently adequate but could be further improved in the future, as is also shown further on in this work (Figure 12, panel b)."*
* * *
**After all the changes that are introduced as replies to your comments/questions #14-18 above, the paragraphs discussing Figures 8 and 9, will be reformatted as follows:**

[revised manuscript text omitted]

      **Reply**: The reviewers suggested that the term "variables" or "parameters" is used instead of "influence quantities". The phrase will be changed to "different variables".

20. Section 4.2, line 366: "But the AMF depends on cloud and surface information?!"

**Reply:** You are correct that the AMF depends on this input information, however it also depends on numerous other variables. In validation studies it is hence always examined on its own accord as well.

21. Figure 10: "Figures changed"
   **Reply**: Panel (a): the only change here is the different color of some data-points to better indicate the solar zenith angle bins with a limited number of co-locations. This information is given in the 1st paragraph of Section 4.2.
   Panel (b): Following the suggestion of one of the reviewers, the viewing zenith angle dependency plot was replaced with the one showing the dependency on the satellite pixel.

22. Section 4.2.2, line 399: "What instrument is this?"
   **Reply**: The phrase will be changed to "that validated their TCWV product against Special Sensor Microwave Imager Sounder (SSMIS) measurements on board f16 and f17,..."

23. Section 4.2.2, line 403: "How different?"
   **Reply**: The differences between the two algorithms will be added in Section 1 (see our answer in your respective question #2, above).

24. Section 4.2.2, line 405: "How is cloud albedo defined?"
   **Reply**: According to https://glossary.ametsoc.org/wiki/Cloud_albedo
   "(Cloud albedo is) The fraction of solar radiation reflected directly by clouds in the atmosphere."
   The reference will be added to the manuscript, even though it is a rather common term.

25. Section 4.2.2, lines 412-414: "If your validation says it's OK, why not use them?" and "I don't understand this."
   **Reply**: As we state in the previous sentences, the co-locations with cloud fraction ranging between 0.3 and 0.5 do not agree well with the ground-based data, having mean relative biases of +20 to 60 %. Nevertheless, they are very few in population so they don't change our validation results, since their contribution is not important. We think that the suggestion to not use these data for scientific studies of the TCWV is within the purposes of a validation paper.
   The sentence will be changed to:
   *"Filtering-out the co-locations dataset for cloud fraction over 0.3 was also investigated, resulting to no major differences in the overall validation results due to the limited contribution of the relatively low number of co-locations with cloud fraction values in the range 0.3 − 0.5. Nevertheless, it is advisable to not use this small portion of TCWV data for future scientific studies"*

26. Section 4.2.2, lines 427-428: "This sounds a bit like "They don't look too good, so we left them out". Either you show them, or you give a scientific argument why not (too few points?)"

**Reply**: The co-locations with surface albedo over 0.2 were left out of the new figure because this was a suggestion of one of the reviewers. She/He proposed to limit the x-axis of the figure up to 0.2 and make the binning of the co-locations with respect to surface albedo finer. We also mention in the same sentence that "very few co-locations have surface albedo above 0.2".

27. Figure 12: *"How come these figures changed so much?"*
    **Reply**:
    Panel (a): The overall number of co-locations changed. Therefore the data-points showing co-locations with surface pressure below 950 hPa, which were few in number in the previous version, became even fewer and changed the statistics of the respective averaging bins.
    Panel (b): The above is valid here, too. Additionally, in this figure the x-axis range was changed as we explained in the previous comment (#26).

28. Competing interests: *"What a strange statement. I mean, can't you just say "D.L. is a member of the editorial board"?"*
    **Reply:** According to the AMT instructions for this field:
    "*If some authors are members of the editorial board of the journal, a sentence should be included for the sake of transparency: "Some authors are members of the editorial board of journal X. The peer-review process was guided by an independent editor, and the authors have also no other competing interests to declare.".*
    We will also add the second part of the statement in our manuscript.